# Data-driven quantum chemical property prediction leveraging 3D conformations with Uni-Mol+

Shuqi Lu [1], Zhifeng Gao [1], Di He [2], Linfeng Zhang[1] & Guolin Ke [1] ✉

Quantum chemical (QC) property prediction is crucial for computational materials and drug design, but relies on expensive electronic structure calculations like density functional theory (DFT). Recent deep learning methods accelerate this process using 1D SMILES or 2D graphs as inputs but struggle to achieve high accuracy as most QC properties depend on refined 3D molecular equilibrium conformations. We introduce `Uni-Mol+`, a deep learning approach that leverages 3D conformations for accurate QC property prediction. `Uni-Mol+` first generates a raw 3D conformation using RDKit then iteratively refines it towards DFT equilibrium conformation using neural networks, which is finally used to predict the QC properties. To effectively learn this conformation update process, we introduce a two-track Transformer model backbone and a novel training approach. Our benchmarking results demonstrate that the proposed `Uni-Mol+` significantly improves the accuracy of QC property prediction in various datasets.

The application of computational methods has become a widely employed strategy in the development of new materials and drugs. A crucial aspect of this approach involves the calculation of quantum chemical (QC) properties of molecular structures[1]. These quantitative properties are highly dependent on the refined equilibrium conformations of molecules.

In the field of materials and drug design, researchers primarily focus on the quantitative properties of equilibrium conformations. The process to achieve this generally involves two key steps, both of which depend on electronic structure methods such as density functional theory (DFT)[2]. The initial step entails performing conformation optimization, also known as energy minimization, on the molecular structure to determine the equilibrium conformation. Subsequently, the quantum chemical (QC) properties of this equilibrium conformation are computed. However, the combined process of conformation optimization and property calculation using DFT can be extremely time-consuming and computationally expensive, potentially requiring several hours to evaluate the properties of just a single molecule. This constraint hinders the applicability of DFT in large-scale data screening endeavors. Consequently, it is of paramount importance to develop alternative methods that maintain the requisite accuracy while reducing computational costs.

Recent studies have demonstrated the potential of using deep learning to accelerate QC property calculations[3–5]. This approach involves training a deep neural network model to predict the property using molecular inputs, thereby circumventing the need for computationally-intensive DFT calculations. Prior research has mainly utilized 1D SMILES[6–8] sequences or 2D molecular graphs[4,9–13] as molecular inputs due to their easy obtainability. However, predicting QC properties from 1D SMILES and 2D molecular graphs can be ineffective since most QC properties are highly related to the refined 3D equilibrium conformations.

To address this challenge, we propose a method called `Uni-Mol+` in this paper, illustrated in Fig. 1a. In contrast to previous approaches that directly predict QC properties from 1D/2D data, `Uni-Mol+` takes advantage of the 3D conformation of the molecule as input, in accordance with physical principles. `Uni-Mol+` first generates a raw 3D conformation from 1D/2D data using cheap methods, such as RDKit[14]. As the raw conformation is inaccurate, `Uni-Mol+` then iteratively updates it towards the DFT equilibrium conformation using neural

[1]DP Technology, Beijing, China. [2]Peking University, Beijing, China. ✉e-mail: kegl@dp.tech

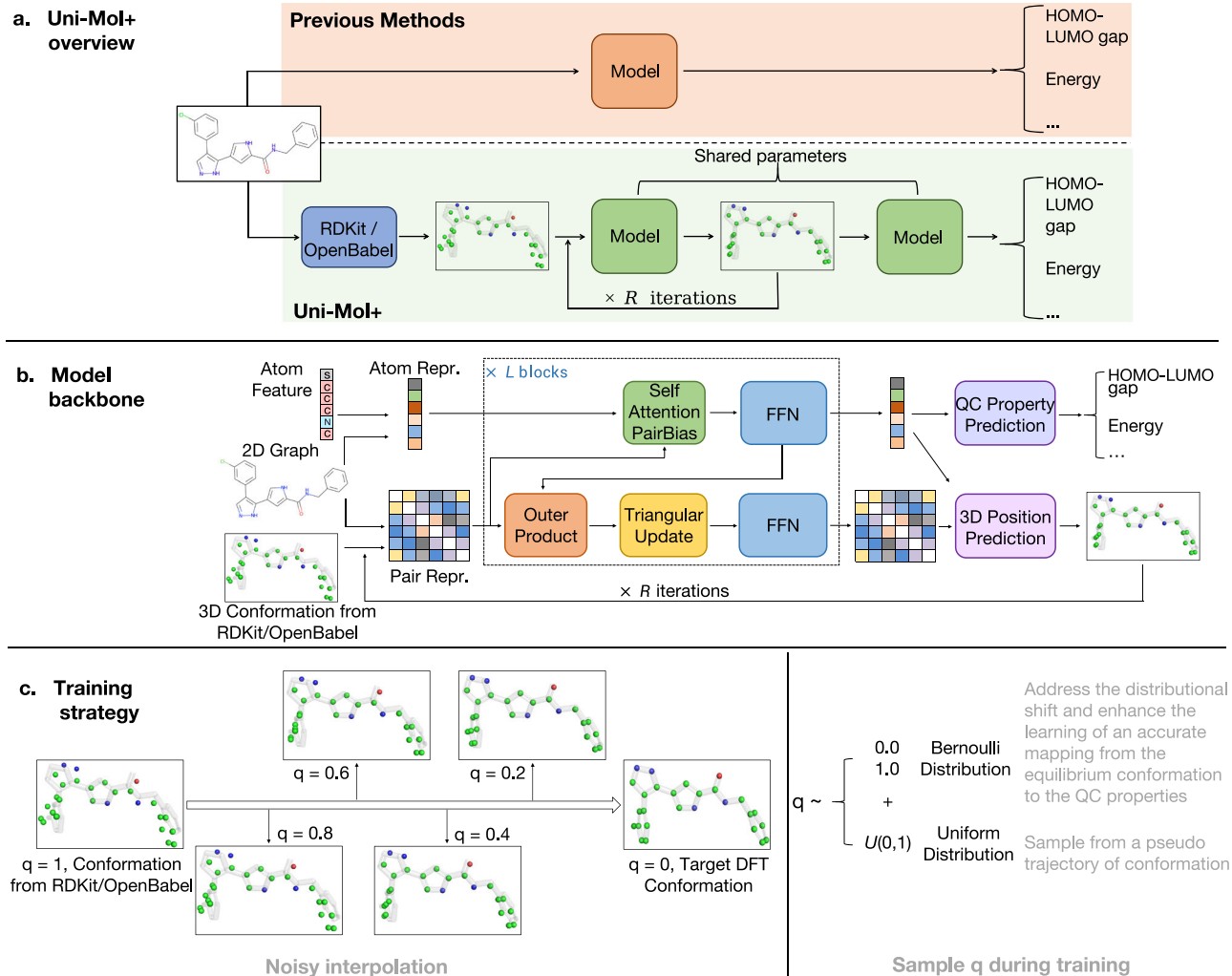

**Fig. 1 | Overall architecture of Uni-Mol+. a** In contrast to prior methods that directly predict QC properties from 1D/2D data, `Uni-Mol+` uses a different approach. It first generates raw 3D conformation from 1D/2D data using cheap tools like RDKit, and then iteratively updates it towards the DFT equilibrium conformation. Finally, it predicts QC properties using the learned conformation. The abbreviation HOMO-LUMO gap represents the Highest Occupied Molecular Orbital − Lowest Unoccupied Molecular Orbital gap. **b** The `Uni-Mol+` backbone consists of $L$ blocks, each of which maintains two tracks of representations−atom and pair, initialized by atom features and 2D graph/3D conformation, respectively. These representations communicate with each other at every block. Based on this backbone model, `Uni-Mol+` iteratively updates the raw conformation (i.e., 3D coordinates of atoms) towards the DFT equilibrium conformation for $R$ iterations. The abbreviation FFN represents the Feed-Forward Neural network and QC property represents Quantum Chemical property. **c** A linear noisy interpolation between raw conformation and DFT conformation is used to generate a pseudo trajectory, effectively augmenting the input conformations. `Uni-Mol+` uses a mixture of Bernoulli distribution and Uniform distribution to sample the noise interpolation weight $q$ during training. The symbol $q$ represents the interpolation weight between raw conformation and DFT conformation.

networks and predicts QC properties from the learned conformation. To obtain accurate equilibrium conformation predictions, we use large-scale datasets (e.g., PCQM4MV2 benchmark) to build up millions of pairs of RDKit-generated raw conformation and high-quality DFT equilibrium conformation and learn the update process from this supervised information. With a carefully designed model backbone and training strategy, `Uni-Mol+` shows superior performance in various benchmarks.

Our main contributions can be summarized as follows:

- We develop a novel paradigm for QC property prediction by leveraging the conformation optimization from RDKit-generated conformation to DFT equilibrium conformation.
- We create a new training strategy for 3D conformation optimization by generating a pseudo trajectory and a sampling strategy from it, based on a mixture of Bernoulli distribution and Uniform distribution.

- The entire framework of `Uni-Mol+` holds significant empirical value, as it achieves markedly better performance than all previous works on two widely recognized benchmarks, PCQM4MV2[15] and Open Catalyst 2020 (OC20)[16].

## Results

In this section, we initially present a concise overview of the `Uni-Mol+` framework, followed by comprehensive benchmarking using two well-recognized public datasets: PCQM4MV2[15] and OC20[16]. These datasets enable the assessment of `Uni-Mol+`'s performance in small organic molecules and catalyst systems. Following this, we perform an ablation study to investigate the impact of various model components and training strategies on the overall performance. Lastly, we present a visual analysis to effectively demonstrate the conformation update process within `Uni-Mol+`. The complete model configuration can be found in the Supplementary Section 2.

## Uni-Mol+ overview

As illustrated in Fig. 1a, for any molecule, `Uni-Mol+` first obtains a raw 3D conformation generated by cheap methods, such as template-based methods from RDKit and OpenBabel. It then learns the target conformation, i.e., the equilibrium conformation optimized by DFT, by an iterative update process from the raw conformation. In the final step, the QC properties are predicted based on the learned conformation. To achieve this goal, we introduce a new model backbone and a novel training strategy for updating conformation and predicting QC properties.

The `Uni-Mol+` 's model backbone is a two-track transformer, consisting of an atom representation track and a pair representation track, as shown in Fig. 1b. In comparison to the transformer backbone used in the prior study Uni-Mol[17], two significant updates have been implemented. i) The pair representation is enhanced by an outer product of the atom representation (referred to `OuterProduct`) for atom-to-pair communication, and a triangular operator (referred to `TriangularUpdate`) to bolster the 3D geometric information. These two operators are proven effective in AlphaFold2[18]. ii) An iterative process is employed to continuously update the 3D coordinates towards the equilibrium conformation. We use $R$ to denote the number of rounds for conformation optimization.

For the learning of the conformation update process, we introduce a novel training strategy as shown in Fig. 1c. We sample conformations from the trajectory between the RDKit-generated raw conformation and the DFT equilibrium conformation, and use the sampled conformation as input to predict the equilibrium conformation. It is crucial to note that the actual trajectory is often unknown in many datasets; therefore, we utilize a pseudo trajectory that presumes a linear process between two conformations. Furthermore, we devise a sampling strategy for obtaining conformations from the pseudo trajectory to serve as the model's input during training. This strategy uses a mixture of Bernoulli distribution and Uniform distribution. The Bernoulli distribution addresses (1) the distributional shift between training and inference and (2) enhances the learning of an accurate mapping from the equilibrium conformation to the QC properties. Meanwhile, the Uniform distribution generates additional intermediate states to serve as model inputs, effectively augmenting the input conformations. The details of `Uni-Mol+` can be found in Sec. 4.

## Benchmark on small molecule (PCQM4MV2)

The PCQM4Mv2 dataset, derived from the OGB Large-Scale Challenge[15], is designed to facilitate the development and evaluation of machine learning models for predicting QC properties of molecules, specifically the target property known as the HOMO-LUMO gap. This property represents the difference between the energies of the highest occupied molecular orbital (HOMO) and the lowest unoccupied molecular orbital (LUMO). The dataset, consisting of approximately 4 million molecules represented by SMILES notations, offers HOMO-LUMO gap labels for the training and validation sets; however, the labels for the test set remain undisclosed. Furthermore, the training set encompasses the DFT equilibrium conformation, which is not included in the validation and test sets. The benchmark's goal is to utilize SMILES notation, without the DFT equilibrium conformation, to predict the HOMO-LUMO gap during the inference process.

Based on SMILES, we generate 8 initial conformations for each molecule by RDKit, at a per-molecule cost of about 0.01 seconds. Specifically, we use ETKDG [19] method to generate 3D conformations. Subsequent optimization of these conformations is achieved through the MMFF94[20] force field. In molecules where the generation of a 3D conformation is unsuccessful, we default to producing a 2D conformation with a flat $z$-axis using RDKit's `AllChem.Compute2DCoords` function instead. During training, we randomly sample 1 conformation as input at each epoch, while during inference, we use the average HOMO-LUMO gap prediction based on 8 conformations.

**Table 1 | The benchmark results on PCQM4MV2**

| Model | # param. | # layers | Valid MAE (↓) | Leaderboard MAE[1] (↓) |
|---|---|---|---|---|
| MLP-Fingerprint[15] | 16.1M | - | 0.1735 | 0.1760 |
| GCN[33] | 2.0M | - | 0.1379 | 0.1398 |
| GIN[34] | 3.8M | - | 0.1195 | 0.1218 |
| GINE-$_{VN}$[3,9,35] | 13.2M | - | 0.1167 | - |
| GCN-$_{VN}$[9,33] | 4.9M | - | 0.1153 | 0.1152 |
| GIN-$_{VN}$[9,34] | 6.7M | - | 0.1083 | 0.1084 |
| DeeperGCN-$_{VN}$[3,36] | 25.5M | 12 | 0.1021 | - |
| GraphGPS$_{SMALL}$[37] | 6.2M | 5 | 0.0938 | - |
| TokenGT[11] | 48.5M | 12 | 0.0910 | 0.0919 |
| GRPE$_{BASE}$[12] | 46.2M | 12 | 0.0890 | - |
| EGT[13] | 89.3M | 24 | 0.0869 | 0.0872 |
| GRPE$_{LARGE}$[13] | 46.2M | 18 | 0.0867 | 0.0876 |
| Graphormer[4,5] | 47.1M | 12 | 0.0864 | - |
| GraphGPS$_{BASE}$[37] | 19.4M | 10 | 0.0858 | - |
| GraphGPS$_{DEEP}$[37] | 13.8M | 16 | 0.0852 | 0.0862 |
| GEM-2[38] | 32.1M | 12 | 0.0793 | 0.0806 |
| GPS++[39] | 44.3M | 16 | 0.0778 | 0.0720[2] |
| Transformer-M[3] | 47.1M | 12 | 0.0787 | - |
| | 69M | 18 | 0.0772 | 0.0782 |
| `Uni-Mol+` | 27.7M | 6 | 0.0714 ±6e−5 | - |
| | 52.4M | 12 | 0.0696 ± 5e−5 | 0.0708 |
| | 77M | 18 | **0.0693 ± 3e−5** | **0.0705** |

[1] The leaderboard was accessed on October 15, 2023, the date of this paper's submission.
[2] GPS++'s leaderboard submission consists of a 112-model ensemble and utilizes the validation data for training.
We highlight the best results in bold. Source data are provided as a Source Data file.

We incorporate previous submissions to the PCQM4MV2 leaderboard as baselines. In addition to the default 12-layer model, we evaluate the performance of `Uni-Mol+` with two variants consisting of 6 and 18 layers, respectively. This aims to explore how model performance changes when varying the model parameter sizes.

The results are summarized in Table 1, and our observations are as follows: (1) `Uni-Mol+` surpasses the previous SOTA by a margin of 0.0079 on validation data on single-model performance, a relative improvement of **11.4%**. (2) All three variants of `Uni-Mol+` demonstrate substantial performance improvements over previous baselines. (3) The 6-layer `Uni-Mol+`, despite having considerably fewer model parameters, outperforms all prior baselines. (4) Increasing the layers from 6 to 12 results in a significant accuracy enhancement, surpassing all baselines by a considerable margin. (5) The 18-layer `Uni-Mol+` exhibits the highest performance, outperforming all baselines by a remarkable margin. These findings underscore the effectiveness of `Uni-Mol+`. (6) The performance of a single 18-layer `Uni-Mol+` model on the leaderboard (test-dev set) is noteworthy, particularly as it surpasses previous state-of-the-art methods without employing an ensemble or additional techniques. In contrast, the previous state-of-the-art GPS++ relied on a 112-model ensemble and included the validation set for training.

## Benchmark on catalyst system (OC20)

The Open Catalyst 2020 (OC20) dataset[16] is specifically designed to promote the development of machine-learning models for catalyst discovery and optimization. OC20 encompasses three tasks: Structure to Energy and Force (S2EF), Initial Structure to Relaxed Structure (IS2RS), and Initial Structure to Relaxed Energy (IS2RE). In this paper, we focus on the IS2RE task, as it aligns well with the objectives of the

**Table 2 | The benchmark results on OC20 IS2RE task**

| | Energy MAE (eV) ↓ | | | | | EwT (%) ↑ | | | | |
|---|---|---|---|---|---|---|---|---|---|---|
| **Results on validation set** | | | | | | | | | | |
| Model | ID | OOD Ads. | OOD Cat. | OOD Both | AVG. | ID | OOD Ads. | OOD Cat. | OOD Both | AVG. |
| SchNet[26] | 0.6465 | 0.7074 | 0.6475 | 0.6626 | 0.6660 | 2.96 | 2.22 | 3.03 | 2.38 | 2.65 |
| DimeNet++[40] | 0.5636 | 0.7127 | 0.5612 | 0.6492 | 0.6217 | 4.25 | 2.48 | 4.40 | 2.56 | 3.42 |
| GemNet-T[41] | 0.5561 | 0.7342 | 0.5659 | 0.6964 | 0.6382 | 4.51 | 2.24 | 4.37 | 2.38 | 3.38 |
| SphereNet[42] | 0.5632 | 0.6682 | 0.5590 | 0.6190 | 0.6024 | 4.56 | 2.70 | 4.59 | 2.70 | 3.64 |
| Graphormer-3D[5] | 0.4329 | 0.5850 | 0.4441 | 0.5299 | 0.4980 | - | - | - | - | - |
| GNS[43] | 0.54 | 0.65 | 0.55 | 0.59 | 0.5825 | - | - | - | - | - |
| GNS+NN[43] | 0.47 | 0.51 | 0.48 | 0.46 | 0.4800 | - | - | - | - | - |
| EquiFormer[44] | 0.4222 | 0.5420 | 0.4231 | 0.4754 | 0.4657 | 7.23 | 3.77 | 7.13 | 4.10 | 5.56 |
| EquiFormer+NN[44] | 0.4156 | 0.4976 | 0.4165 | 0.4344 | 0.4410 | 7.47 | 4.64 | 7.19 | 4.84 | 6.04 |
| DRFormer[45] | 0.4187 | 0.4863 | 0.4321 | 0.4332 | 0.4425 | 8.39 | 5.42 | 8.12 | 5.44 | 6.84 |
| Uni-Mol+ | **0.3787** | **0.4519** | **0.4009** | **0.4048** | **0.4119** | **11.02** | **6.61** | **10.00** | **6.38** | **8.60** |
| | ± 0.0007 | ± 0.0049 | ± 0.0001 | ± 0.0037 | ± 0.0036 | | | | | |
| **Results on test set** | | | | | | | | | | |
| SchNet[26] | 0.639 | 0.734 | 0.662 | 0.704 | 0.6848 | 2.96 | 2.33 | 2.94 | 2.21 | 2.61 |
| DimeNet++[40] | 0.562 | 0.725 | 0.576 | 0.661 | 0.631 | 4.25 | 2.07 | 4.1 | 2.41 | 3.21 |
| SphereNet[42] | 0.563 | 0.703 | 0.571 | 0.638 | 0.6188 | 4.47 | 2.29 | 4.09 | 2.41 | 3.32 |
| Graphormer-3D[5] | 0.3976 | 0.5719 | 0.4166 | 0.5029 | 0.4722 | 8.97 | 3.45 | 8.18 | 3.79 | 6.1 |
| GNS+NN[43] | 0.4219 | 0.5678 | 0.4366 | 0.4651 | 0.4728 | 9.12 | 4.25 | 8.01 | 4.64 | 6.5 |
| EquiFormer[44] | 0.5037 | 0.6881 | 0.5213 | 0.6301 | 0.5858 | 5.14 | 2.41 | 4.67 | 2.69 | 3.73 |
| EquiFormer+NN[44] | 0.4171 | 0.5479 | 0.4248 | 0.4741 | 0.4660 | 7.71 | 3.70 | 7.15 | 4.07 | 5.66 |
| DRFormer[45] | 0.3865 | 0.5435 | 0.4060 | 0.4677 | 0.4509 | 9.18 | 4.01 | 8.39 | 4.33 | 6.48 |
| Uni-Mol+ | **0.3745** | **0.4760** | **0.3980** | **0.4086** | **0.4143** | **11.29** | **6.05** | **9.53** | **6.06** | **8.23** |

NN refers to Noisy Nodes[43]. We highlight the best results in bold. Source data are provided as a Source Data file.

proposed methodology. The goal of the IS2RE task is to predict the relaxed energy based on the initial conformation. It comprises approximately 460K training data points. While DFT equilibrium conformations are provided for training, they are not permitted for use during inference. Moreover, in contrast to the PCQM4MV2 dataset, the initial conformation is already supplied in the OC20 IS2RE task, eliminating the need to generate the initial input conformation by ourselves.

We present a performance comparison of various models on the OC20 IS2RE validation and test set, as illustrated in Table 2. The table displays the Mean Absolute Error (MAE) for energy in electron volts (eV) and the percentage of Energies Within a Threshold (EwT) for each model. As evident from the tables, our proposed Uni-Mol+ significantly outperforms all previous baselines in terms of both MAE and EwT. For example, in the test set, Uni-Mol+ exceeds the previous SOTA in Average MAE and Average EwT by margins of 0.0366 (**8.8%** relative improvement) and 1.73 (**26.6%** relative improvement), respectively. This demonstrates the exceptional performance of Uni-Mol+. Notably, our method attains the lowest MAEs across all categories, including In-Domain (ID), Out-of-Domain Adsorption (OOD Ads.), Out-of-Domain Catalysis (OOD Cat.), Out-of-Domain Both (OOD Both), and Average (AVG.). Furthermore, in terms of EwT, Uni-Mol+ consistently achieves the highest values in all categories. These findings underscore the robustness of our method in handling both in-domain and out-of-domain data. In conclusion, the results emphasize the efficacy of our approach in capturing intricate interactions in material systems and its potential for extensive applicability in various computational material science tasks.

**Ablation study**

In this subsection, we present a comprehensive ablation study for Uni-Mol+. To fully comprehend the configurations discussed herein,

we recommend referring to the "Methods" section and the model specifications detailed in Supplementary Section 2. We conduct the ablation study on the PCQM4Mv2 dataset, employing the default 12-layer Uni-Mol+ configuration. The findings are summarized in Table 3, where No. 1 is the default setting, and No.2–7 focus on the examination of the model backbone, and No. 8 to No. 17 focus on the examination of the training strategies. A detailed analysis follows in the subsequent paragraphs.

As detailed in Sec. 4 and Supplementary Section 1, Uni-Mol+ introduces two novel components, OuterProduct and TriangularUpdate, and iteratively updates the 3D coordinates. An examination of the results (No. 1–7) in Table 3 provides insights into the implications of these modifications.

(1) We first examine the necessity of the new components in the model backbone. Upon examining the first three settings (No. 1 to 3), it becomes evident that both TriangularUpdate and OuterProduct significantly contribute to the model's performance. A comparison between No. 3 and No. 4 reveals that utilizing pair representation exclusively, without incorporating OuterProduct or TriangularUpdate, does not enhance performance. This result is expected because the pair representation is not communicated with the atom representation (without OuterProduct) and is simply updated by FFN, resulting in a performance that is almost the same as not using pair representation, as there are merely more parameters. However, the proposed OuterProduct and TriangularUpdate can better utilize the pair representation, leading to an overall performance improvement (No.1 and No.2). This makes the pair representation an essential component in the backbone of our approach, even if its standalone effectiveness might appear limited.

(2) We then examine the performance brought by iterative coordinate updates. A comparison of No. 1 with No. 5 and No. 6 leads to the

**Table 3 | Ablation study for model backbone and for sampling strategies for $q$, on PCQM4MV2**

| No. | TriangularUpdate | OuterProduct | Pair Repr. | $R$ | $w_{1.0}$ | $w_{0.0}$ | $w_u$ | Valid MAE (↓) |
|---|---|---|---|---|---|---|---|---|
| 1 | ✓ | ✓ | ✓ | 1 | 0.1 | 0.8 | 0.1 | <u>0.0696</u> |
| Ablation study on model backbone | | | | | | | | |
| 2 | ✗ | ✓ | ✓ | 1 | 0.1 | 0.8 | 0.1 | 0.0704 |
| 3 | ✗ | ✗ | ✓ | 1 | 0.1 | 0.8 | 0.1 | 0.0710 |
| 4 | ✗ | ✗ | ✗ | 1 | 0.1 | 0.8 | 0.1 | 0.0709 |
| 5 | ✓ | ✓ | ✓ | 0 | 0.1 | 0.8 | 0.1 | 0.0715 |
| 6 | ✓ | ✓ | ✓ | 2 | 0.1 | 0.8 | 0.1 | **0.0695** |
| 7 | ✗ | ✗ | ✗ | 0 | 0.1 | 0.8 | 0.1 | 0.0738 |
| Ablation study on training strategy | | | | | | | | |
| 8 | ✓ | ✓ | ✓ | 1 | 1.0 | - | - | 0.0771 |
| 9 | ✓ | ✓ | ✓ | 1 | - | 1.0 | - | 0.1122 |
| 10 | ✓ | ✓ | ✓ | 1 | - | - | 1.0 | 0.0724 |
| 11 | ✓ | ✓ | ✓ | 1 | 0.1 | 0.9 | - | 0.0697 |
| 12 | ✓ | ✓ | ✓ | 1 | - | 0.9 | 0.1 | 0.0753 |
| 13 | ✓ | ✓ | ✓ | 1 | 0.1 | 0.7 | 0.2 | 0.0698 |
| 14 | ✓ | ✓ | ✓ | 1 | 0.2 | 0.7 | 0.1 | 0.0703 |
| 15 | ✓ | ✓ | ✓ | 1 | 0.1 | 0.6 | 0.3 | 0.0702 |
| 16 | ✓ | ✓ | ✓ | 1 | 0.2 | 0.6 | 0.2 | 0.0706 |
| 17 | ✓ | ✓ | ✓ | 1 | 0.3 | 0.6 | 0.1 | 0.0714 |
| 18 | ✓ | ✓ | ✓ | 1 | Noisy Nodes | | | 0.0760 |
| 19 | ✗ | ✗ | ✗ | 1 | Noisy Nodes | | | 0.0798 |

$R$ refers to the number of rounds of conformation updates. $w_{1.0}$ refers to the sample probability of RDKit conformation, $w_{0.0}$ refers to the sample probability of target conformation with noise and $w_u$ refers to the sa probability of intermediate conformation. We highlight the best results in bold. We use underlines to indicate the results under the standard settings. Source data are provided as a Source Data file.

conclusion that omitting the iterative update (No. 5) yields suboptimal results. Note that even without the iterative refinement of 3D conformation (R = 0), Uni-Mol+ 's score of 0.0715 (No. 5) significantly surpasses the previous SOTA GPS++ (0.0778). However, performing one additional iteration proves highly effective (No. 1), whereas further increasing the number of iterations offers marginal improvements (No. 6).

(3) Lastly, we check the result using the same model backbone as previous work. In particular, when the model retains the same structure as the one employed in previous works[3,5,17] and excludes the iterative update (No. 7), its performance is the least favorable. Nonetheless, even with this substandard performance, the model surpasses all prior baselines, thereby highlighting the efficacy of the proposed training strategy. It is important to note that No.7 employs the proposed training strategy as outlined in Sec. 4.2. Although No.7 does not explicitly use conformation optimization (R = 0), the model is still trained to predict the target conformation. Consequently, the Atom Repr. and Pair Repr. of the last layer inherently contain the information required to predict the target conformation. Hence, even without explicitly conformation optimization (R=0), the result of No.7 still supports our primary contribution, namely the accurate prediction of QC properties by leveraging an auxiliary task of conformation optimization.

The training strategy primarily concentrates on sampling $q$ (interpolation weight, details in Sec. 4) to obtain input conformations during training. Formally, $q$ is sampled from a mixture of Bernoulli and Uniform distributions, denoted as $w_{1.0}\mathbb{I}_{\{1.0\}}(q) + w_{0.0}\mathbb{I}_{\{0.0\}}(q) + w_u \mathbb{I}_{[a,b]}(q)$, where $\mathbb{I}_{\{c\}}(q)$ is an indicator function that equals 1 if $q = c$ and 0 otherwise, and $\mathbb{I}_{[a,b]}(q)$ is an indicator function that equals 1 if $a \le q \le b$ and 0 otherwise. The weights $w_{1.0}$, $w_{0.0}$, and $w_u$ must be non-negative and add up to 1, i.e., $w_{1.0} + w_{0.0} + w_u = 1$. In this notation, the default sampling strategy employed in Uni-Mol+ can be represented as $(w_{1.0} = 0.1, w_{0.0} = 0.8, w_u = 0.1, [a, b] = [0.4, 0.6])$. We investigate additional settings for the ablation study, and the results are summarized in

Table 3 (No. 8 to No. 17). Except for No. 10 and 12, which use $[a, b] = [0.0, 1.0]$, all other settings use $[a, b] = [0.4, 0.6]$. From these results, we make the following observations:

(1) Comparing No. 8, 9, and 10, we find that sampling from only one type of conformation is not effective. For No. 8, it lacks data augmentation and cannot learn an accurate mapping from equilibrium conformation to QC property. For No. 9, it experiences a distributional shift between training and inference. Although No. 10 is better, it has a low probability of sampling 0.0 and 1.0, resulting in suboptimal performance.

(2) By comparing No. 8, 9, and 11, we can deduce that sampling from the mixture of RDKit and target conformations yields a satisfactory result (Valid MAE with 0.0697). However, if only sampling from target and intermediate conformations (No. 12), the result is unsatisfactory (Valid MAE with 0.0753). This result indicates that sampling from $w_{1.0}$ is necessary, as it reduces the distributional shift between training and inference.

(3) The default strategy that samples from three types of conformations (No. 1) exhibits the best performance.

(4) Altering the weights of the mixture distribution (No. 13–17) does not result in better performance over the default strategy. Furthermore, we notice that with a decreased $w_{0.0}$, the performance worsens. This suggests that the default weighting scheme is appropriate for this task.

(5) Upon comparing the results of No.18 and No.1, it's clear that the performance of Noisy Nodes (No.18, Valid MAE with 0.0760) is significantly lower than that of Uni-Mol+ (No.1, Valid MAE with 0.0696). This large performance gap (0.0760 vs. 0.0696) highlights the superior efficacy of the proposed training strategy, as opposed to the one employed previously.

(6) A comparison between No.19 and No.18 shows that the model structure employed in previous works[3,5,17] yields worse results than using Uni-Mol+ 's backbone when using Noisy Nodes strategy. This finding lends additional support to the superiority of

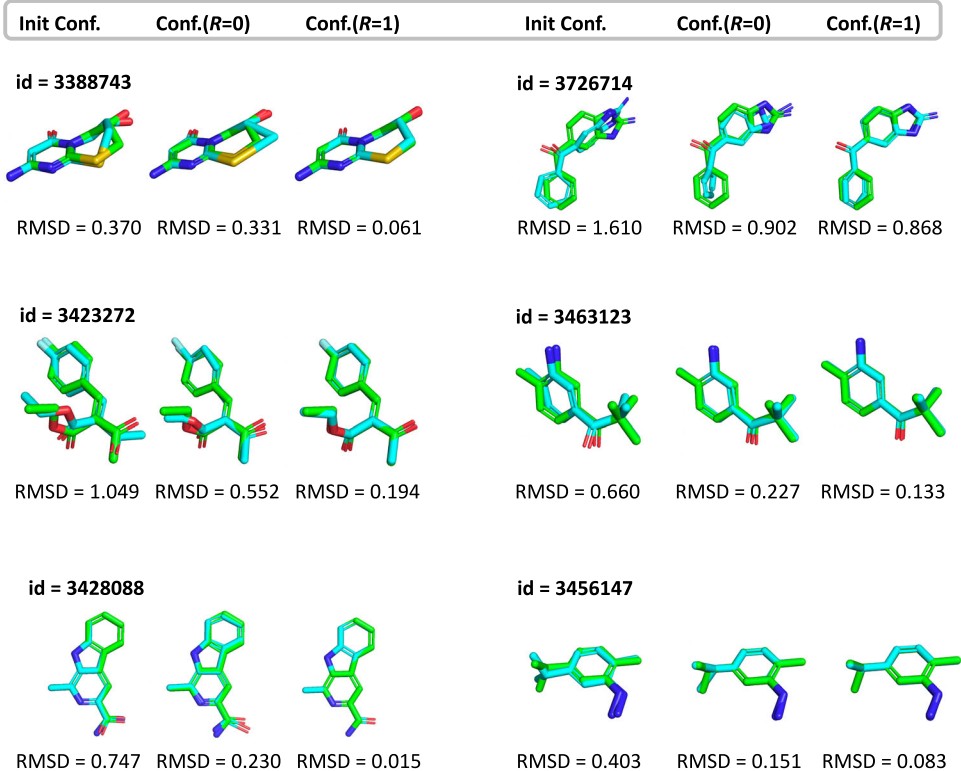

**Fig. 2 | Visualization of** `Uni-Mol+` **'s predicted conformations.** Comparison of RDKit-generated conformation and predicted conformations from first ($R=0$) and second ($R=1$) iterations, superimposed onto the target DFT conformation. Corresponding RMSDs are provided, demonstrating `Uni-Mol+` 's effectiveness in predicting accurate DFT equilibrium conformations. The abbreviations RMSD represents Root Mean Square Deviation. The conformations are provided in the Supplementary Data 1.

`Uni-Mol+` 's backbone over the model architectures previously proposed.

In conclusion, the ablation study demonstrates the effectiveness of the default sampling strategy employed in `Uni-Mol+`, emphasizing the importance of utilizing a mixture of different conformations to achieve superior performance.

### Visualized analysis of conformation learning

In addition to QC property prediction, `Uni-Mol+` can also predict equilibrium conformations. Although this study primarily focuses on QC property prediction and the previous experimental results have clearly demonstrated the effectiveness of the proposed `Uni-Mol+`, visualized results can help to better understand how `Uni-Mol+` works. Therefore, we also provide two additional analyzes for the conformation learning of `Uni-Mol+` in the PCQM4MV2 dataset.

The First analysis evaluates the predicted conformations. Since the DFT conformations of the validation set (and test set) are not provided by the PCQM4MV2 dataset, we generated DFT conformations ourselves, using the same settings as the PCQM4MV2 source data[21]. As shown in Fig. 2, `Uni-Mol+` can effectively predict equilibrium conformations. Moreover, as the number of update iterations increases, the RMSD is smaller, further demonstrating the effectiveness of the proposed iterative coordinate update. We provide the conformation files used in Fig. 2 in Supplementary Data 1.

The second analysis aims to show that `Uni-Mol+` can predict conformations with lower energies, which approaches equilibrium conformations. To demonstrate this, we selected 100 data points and calculated the energies of their initial and predicted conformations and that between their initial conformations and the DFT conformations. Here the DFT conformations is Computed by ourself using the B3LYP functional and 6-31G* basis set, consistent with the settings used in the PCQM4MV2 dataset. As shown in Fig. 3, `Uni-Mol+` can predict

the conformations with lower energies. Moreover, the energy difference distribution between the initial and predicted conformations closely aligns with that between the initial and equilibrium conformations. This similarity demonstrates `Uni-Mol+` 's effectiveness in predicting equilibrium conformations accurately. We provide the conformation files used in Fig. 3 in Supplementary Data 1.

The aforementioned results provide additional evidence of the effectiveness of the proposed `Uni-Mol+`, as it can indeed predict conformations with lower energy and iteratively approach the target DFT conformations.

## Discussion

Previous studies have primarily relied on 1D/2D information, such as SMILES or molecular graphs, for making predictions[6–9]. Recently, numerous investigations[4,9–13] have employed Transformer models for graph tasks, resulting in significant advancements. Given the importance of 3D information in predicting quantum chemistry (QC) properties, several recent studies have incorporated 3D data into their approaches.

Some research has utilized 3D structural information and maximised mutual information between 2D and 3D molecular to augment 2D representations during training[3,22–24]. However, these studies only implicitly embed 3D information into 2D representations, with 2D data utilized exclusively during inference. We represent these models as $x_{2D} \rightarrow (x_{3D}, y)$, where $x_{2D}$ represents the 2D molecular graph input, $x_{3D}$ represents the 3D conformation input and $y$ denotes a QC property. A crucial shortcoming of these approaches is that they don't explicitly learn a mapping from the 3D equilibrium conformation $x_{3D}$ to $y$ while $y$ is highly correlated with $x_{3D}$. Some models, like Transformer-M[3], attempt to learn both $x_{2D} \rightarrow y$ and $x_{3D} \rightarrow y$. However, during inference, these models rely solely on $x_{2D}$, which compromises the prediction performance. `Uni-Mol+`, on the other hand, employs a strategy

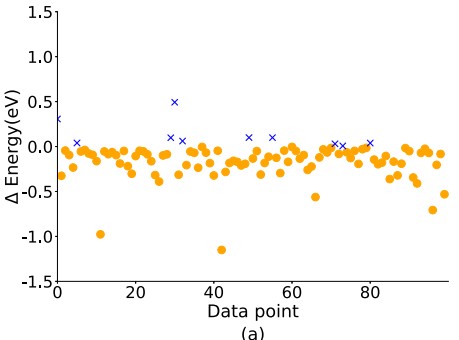

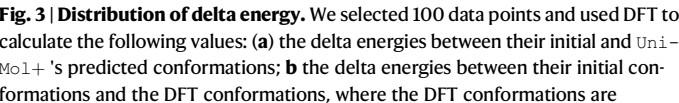

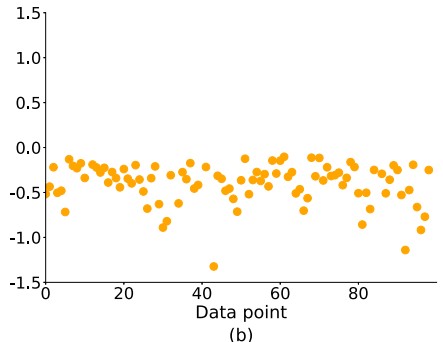

**Fig. 3 | Distribution of delta energy.** We selected 100 data points and used DFT to calculate the following values: (**a**) the delta energies between their initial and Uni-Mol+ 's predicted conformations; **b** the delta energies between their initial conformations and the DFT conformations, where the DFT conformations are calculated by ourselves using DFT tool. Cross-marks indicate data points with increased energies, while circle-marks denote those with decreased energies. This visualization demonstrates that Uni-Mol+ effectively predicts conformations with lower energies. The conformations are provided in the Supplementary Data 1.

$x'_{3D} \rightarrow \dots \rightarrow x_{3D} \rightarrow y$. This process starts with a raw 3D conformation $x'_{3D}$, iteratively refines it towards $x_{3D}$, and then predicts $y$. By explicitly learning a mapping from 3D conformation to QC properties, Uni-Mol + proves to be more effective than previous models.

A few recent works have focused on property prediction using 3D conformations as input. For example, Uni-Mol[17] employs the 3D conformation generated by RDKit as input. Uni-Mol is a pre-training method centred on designing pre-text tasks for molecular data, while Uni-Mol+ is a supervised learning approach aimed at predicting QC properties from raw conformations, aided by equilibrium conformation during training. Graphormer-3D[5] utilizes the initial 3D conformation provided by the OC20 dataset[16] to predict energy at equilibrium. However, it focuses on directly learning the mapping from input to target conformations without considering a training strategy specifically tailored for conformation optimization, as done in our work. The Noisy Nodes approach[25] takes corrupted DFT conformations as inputs and aims to predict the uncorrupted ones. When an initial 3D conformation is provided, as in the OC20 dataset, Noisy Nodes generates an interpolated conformation between the initial and target conformations during training, which is similar to the uniform sampling of $q$ in our study. In comparison to Noisy Nodes, our training strategy also incorporates a Bernoulli distribution, which has proven advantageous in addressing distributional shifts and improving QC property predictions. Moreover, both Graphformer-3D and Noisy Nodes necessitate the use of initial conformations provided by the dataset. In contrast, our study is not constrained by this requirement, as it can employ RDKit to generate initial conformations. Several studies[26–29] concentrate on designing new model backbones with rotation and translation equivalence or invariance in 3D space. In contrast, our work emphasizes a novel paradigm for QC property prediction, rather than developing a new model backbone.

Conformation optimization is a critical challenge in computational chemistry. Density Functional Theory (DFT) is the most prevalent method for this task, offering high accuracy but at considerable computational expense. Several deep learning-based potential energy models, such as Deep Potential[30], have been proposed to tackle this issue by using neural networks to replace costly potential calculations in DFT, thereby enhancing efficiency. However, deep potential models still necessitate dozens or even hundreds of iterative steps to optimize the conformation based on predicted potentials. In contrast, our approach, Uni-Mol+, requires only a few optimization rounds and can optimize conformations end-to-end, whereas deep potential models cannot.

Although other studies[17,31] also optimize RDKit-generated conformations towards DFT conformations, they primarily focus on benchmarking conformation rather than predicting QC property. These works simply employ existing model backbones and learn the mapping between raw and equilibrium conformations. In contrast, Uni-Mol+ adopts a novel training strategy to effectively learn conformation optimization. However, it is important to note that conformation optimization serves merely as an auxiliary task; the primary objective of Uni-Mol+ is to predict QC properties.

The research most closely related to ours is EMPNN[32], which utilizes a 2D molecular graph as input for predicting the 3D equilibrium conformation. However, EMPNN learns to map a 2D graph to a 3D equilibrium conformation, which differs from our model that optimizes from an RDKit-generated conformation. Moreover, EMPNN requires an additional model, such as SchNet[26], to predict quantum chemistry (QC) properties using the 3D conformation generated by EMPNN as input.

In summary, our study presents a novel method capable of accurately predicting QC properties through an auxiliary task of conformation optimization. This approach has the potential to enhance the efficiency of high-throughput screening and facilitate the design of innovative materials and molecules in future research.

## Method
### Model backbone
The designed model backbone can predict the equilibrium conformation and QC property simultaneously, denoted as $(y, \hat{r}) = f(X, E, r; \theta)$. The model takes three inputs, (i) atom features ($X \in \mathbb{R}^{n \times d_f}$, where $n$ is the number of atoms and $d_f$ is atom feature dimension), (ii) edge features ($E \in \mathbb{R}^{n \times n \times d_e}$, where $d_e$ is the edge feature dimension), and (iii) 3D coordinates of atoms ($r \in \mathbb{R}^{n \times 3}$). $\theta$ is the set of learnable parameters. And the model predicts a quantum property $y$ and updated 3D coordinates $\hat{r} \in \mathbb{R}^{n \times 3}$.

As illustrated in Fig. 1b, the $L$-block model maintains two distinct representation tracks: atom representation and pair representation. The atom representation is denoted as $x \in \mathbb{R}^{n \times d_x}$, where $d_x$ represents the dimension of the atom representation. Similarly, the pair representation is denoted as $p \in \mathbb{R}^{n \times n \times d_p}$, where $d_p$ signifies the dimension of the pair representation. The model comprises $L$ blocks, with $x^{(l)}$ and $p^{(l)}$ representing the output representations of the $l$-th block. Within each block, the atom representation is initially updated through self-attention, incorporating an attention bias derived from the pair representation, followed by an update via a feed-forward network (FFN). Concurrently, the pair representation undergoes a series of updates, beginning with an outer product of the atom representation (referred to OuterProduct), followed by triangular multiplication (referred to TriangularUpdate) as implemented in AlphaFold2[18], and finally, an update using a FFN. This backbone, in comparison to the one used in Uni-Mol[17], enhances the pair representation through two key improvements: (i) employing an outer product for effective atom-to-pair communication, and (ii) utilizing a triangular operator to

bolster the 3D geometric information. Next we will introduce each module in detail.

**Positional encoding.** Similar to previous works[4,17], we use pair-wise encoding to encode the 3D spatial and 2D graph positional information. Specifically, for 3D spatial information, we utilize the Gaussian kernel for encoding, as done in previous studies[5,17]. The encoded 3D spatial positional encoding is denoted by $\psi^{3D}$.

In addition to the 3D positional encodings, we also incorporate graph positional encodings similar to those used in Graphormer. This includes the shortest-path encoding, represented by $\psi_{i,j}^{SP} = \text{Embedding}(\text{sp}_{ij})$ where $\text{sp}_{ij}$ is the shortest path between atoms $(i, j)$ in the molecular graph. Additionally, instead of the time-consuming multi-hop edge encoding method used in Graphormer, we utilize a more efficient one-hop bond encoding, denoted by $\psi^{Bond} = \sum_{i=1}^{d_e} \text{Embedding}(\boldsymbol{E}_i)$, where $\boldsymbol{E}_i$ is the $i$-th edge feature. Combined above, the positional encoding is denoted as $\psi = \psi^{3D} + \psi^{SP} + \psi^{Bond}$. And the pair representation $\boldsymbol{p}$ is initialized by $\psi$, i.e., $\boldsymbol{p}^{(0)} = \psi$.

**Update of atom representation.** The atom representation $\boldsymbol{x}^{(0)}$ is initialized by the embeddings of atom features, the same as Graphormer. At $l$-th block, $\boldsymbol{x}^{(l)}$ is sequentially updated as follow:

$$\boldsymbol{x}^{(l)} = \boldsymbol{x}^{(l-1)} + \text{SelfAttentionPairBias}\left(\boldsymbol{x}^{(l-1)}, \boldsymbol{p}^{(l-1)}\right),$$
$$\boldsymbol{x}^{(l)} = \boldsymbol{x}^{(l)} + \text{FFN}\left(\boldsymbol{x}^{(l)}\right). \tag{1}$$

The SelfAttentionPairBias function is denoted as:

$$\boldsymbol{Q}^{(l,h)} = \boldsymbol{x}^{(l-1)} \boldsymbol{W}_Q^{(l,h)}; \boldsymbol{K}^{(l,h)} = \boldsymbol{x}^{(l-1)} \boldsymbol{W}_K^{(l,h)};$$
$$\boldsymbol{B}^{(l,h)} = \boldsymbol{p}^{(l-1)} \boldsymbol{W}_B^{(l,h)}; \boldsymbol{V}^{(l,h)} = \boldsymbol{x}^{(l-1)} \boldsymbol{W}_V^{(l,h)};$$
$$\text{output} = \text{softmax}\left(\frac{\boldsymbol{Q}^{(l,h)}(\boldsymbol{K}^{(l,h)})^T}{\sqrt{d_h}} + \boldsymbol{B}^{(l,h)}\right)\boldsymbol{V}^{(l,h)}; \tag{2}$$

where $d_h$ is the head dimension, $\boldsymbol{W}_Q^{(l,h)}, \boldsymbol{W}_K^{(l,h)}, \boldsymbol{W}_V^{(l,h)} \in \mathbb{R}^{d_x \times d_h}$, $\boldsymbol{W}_B^{(l,h)} \in \mathbb{R}^{d_p \times 1}$. FFN is a feed-forward network with one hidden layer. For simplicity, layer normalizations are omitted. Compared to the standard Transformer layer, the only difference here is the usage of attention bias term $\boldsymbol{B}^{(l, h)}$ to incorporate $\boldsymbol{p}^{(l-1)}$ from the pair representation track.

**Update of pair representation.** The pair representation $\boldsymbol{p}^{(0)}$ is initialized by the positional encoding $\psi$. The update process of pair representation begins with an outer product of $\boldsymbol{x}^{(l)}$, followed by a $\mathcal{O}(n^3)$ triangular multiplication, and is then concluded with an FFN layer. Formally, at $l$-th block, $\boldsymbol{p}^{(l)}$ is sequentially updated as follow:

$$\boldsymbol{p}^{(l)} = \boldsymbol{p}^{(l-1)} + \text{OuterProduct}(\boldsymbol{x}^{(l)});$$
$$\boldsymbol{p}^{(l)} = \boldsymbol{p}^{(l)} + \text{TriangularUpdate}(\boldsymbol{p}^{(l)}); \tag{3}$$
$$\boldsymbol{p}^{(l)} = \boldsymbol{p}^{(l)} + \text{FFN}(\boldsymbol{p}^{(l)}).$$

The OuterProduct is used for atom-to-pair communication, denoted as :

$$\boldsymbol{a} = \boldsymbol{x}^{(l)} \boldsymbol{W}_{O1}^{(l)}, \boldsymbol{b} = \boldsymbol{x}^{(l)} \boldsymbol{W}_{O2}^{(l)};$$
$$\boldsymbol{o}_{i,j} = \text{flatten}(\boldsymbol{a}_i \otimes \boldsymbol{b}_j); \tag{4}$$
$$\text{output} = \boldsymbol{o}\boldsymbol{W}_{O3}^{(l)},$$

where $\boldsymbol{W}_{O1}^{(l)}, \boldsymbol{W}_{O2}^{(l)} \in \mathbb{R}^{d_x \times d_o}$, $d_o$ is the hidden dimension of OuterProduct, and $\boldsymbol{W}_{O3}^{(l)} \in \mathbb{R}^{d_o^2 \times d_p}$, $\boldsymbol{o} = [\boldsymbol{o}_{i,j}]$. Please note that $\boldsymbol{a}, \boldsymbol{b}, \boldsymbol{o}$ are temporary variables in the OuterProduct function.

TriangularUpdate is used to enhance pair representation further, denoted as:

$$\boldsymbol{a} = \text{sigmoid}\left(\boldsymbol{p}^{(l)} \boldsymbol{W}_{T1}^{(l)}\right) \odot \left(\boldsymbol{p}^{(l)} \boldsymbol{W}_{T2}^{(l)}\right);$$
$$\boldsymbol{b} = \text{sigmoid}\left(\boldsymbol{p}^{(l)} \boldsymbol{W}_{T3}^{(l)}\right) \odot \left(\boldsymbol{p}^{(l)} \boldsymbol{W}_{T4}^{(l)}\right);$$
$$\boldsymbol{o}_{i,j} = \sum_k \boldsymbol{a}_{i,k} \odot \boldsymbol{b}_{j,k} + \sum_k \boldsymbol{a}_{k,i} \odot \boldsymbol{b}_{k,j}; \tag{5}$$
$$\text{output} = \text{sigmoid}\left(\boldsymbol{p}^{(l)} \boldsymbol{W}_{T5}^{(l)}\right) \odot \left(\boldsymbol{o}\boldsymbol{W}_{T6}^{(l)}\right),$$

where $\boldsymbol{W}_{T1}^{(l)}, \boldsymbol{W}_{T2}^{(l)}, \boldsymbol{W}_{T3}^{(l)}, \boldsymbol{W}_{T4}^{(l)} \in \mathbb{R}^{d_p \times d_t}$, $\boldsymbol{W}_{T5}^{(l)} \in \mathbb{R}^{d_p \times d_p}$, $\boldsymbol{W}_{T6}^{(l)} \in \mathbb{R}^{d_t \times d_p}$, $\boldsymbol{o} = [\boldsymbol{o}_{i,j}]$, and $d_t$ is the hidden dimension of TriangularUpdate. $\boldsymbol{a}, \boldsymbol{b}, \boldsymbol{o}$ are temporary variables. The TriangularUpdate is inspired by the Evoformer in AlphaFold2[18]. The difference is that AlphaFold2 uses two modules, "outgoing" ($\boldsymbol{o}_{i, j} = \sum_k \boldsymbol{a}_{i,k} \odot \boldsymbol{b}_{j,k}$) and "incoming" ($\boldsymbol{o}_{i,j} = \sum_k \boldsymbol{a}_{k,i} \odot \boldsymbol{b}_{k,j}$) respectively. In Uni-Mol+, we merge the two modules into one to save the computational cost.

**Conformation optimization.** The conformation optimization process in many practical applications, such as Molecular Dynamics, is iterative. This approach is also employed in the Uni-Mol+. The number of conformation update iterations denoted as $R$, is a hyperparameter. We use superscripts on $\boldsymbol{r}$ to distinguish the 3D positions of atoms in different iterations. for example, at the $i$-th iteration, the update can be denoted as $(y, \boldsymbol{r}^{(i)}) = f(\boldsymbol{X}, \boldsymbol{E}, \boldsymbol{r}^{(i-1)}; \theta)$. It is noteworthy that parameters $\theta$ are shared across all iterations. Moreover, please note that the iterative update in Uni-Mol+ involves only a few rounds, such as 1 or 2, instead of dozens or hundreds of steps in Molecular Dynamics.

**3D position prediction head.** Regarding the 3D position prediction head within Uni-Mol+, we have adopted the 3D prediction head proposed in Graphormer-3D[5], as cited in our manuscript. The architecture takes atom representation $\boldsymbol{x}^L$, pair representation $\boldsymbol{p}^L$, and initial coordinates $\boldsymbol{c}$ as inputs. An attention mechanism is initially employed and then the attention weights is multiplied point-wisely with the pairwise delta coordinates derived from the initial coordinates. Similar to SelfAttentionPairBias, the attention mechanism is denoted as:

$$\boldsymbol{Q}^h = \boldsymbol{x}^L \boldsymbol{W}_Q^h; \boldsymbol{K}^h = \boldsymbol{x}^L \boldsymbol{W}_K^h;$$
$$\boldsymbol{B}^h = \boldsymbol{p}^L \boldsymbol{W}_B^h; \boldsymbol{V}^h = \boldsymbol{x}^L \boldsymbol{W}_V^h;$$
$$\boldsymbol{A}_{i,j}^h = \text{softmax}\left(\frac{\boldsymbol{Q}_i^h(\boldsymbol{K}_j^h)^T}{\sqrt{d_h}} + \boldsymbol{B}_{i,j}^h\right); \tag{6}$$
$$\boldsymbol{c}_{i,j} = \boldsymbol{c}_i - \boldsymbol{c}_j; \boldsymbol{A}_{i,j}^{(h,0)} = \boldsymbol{A}_{i,j}^h \odot \Delta\boldsymbol{c}_{i,j}^0;$$
$$\boldsymbol{A}_{i,j}^{(h,1)} = \boldsymbol{A}_{i,j}^h \odot \Delta\boldsymbol{c}_{i,j}^1; \boldsymbol{A}_{i,j}^{(h,2)} = \boldsymbol{A}_{i,j}^h \odot \Delta\boldsymbol{c}_{i,j}^2;$$

where $d_h$ is the head dimension, $\boldsymbol{W}_Q^h, \boldsymbol{W}_K^h, \boldsymbol{W}_V^h \in \mathbb{R}^{d_x \times d_h}$, $\boldsymbol{W}_B^h \in \mathbb{R}^{d_p \times 1}$. $\boldsymbol{A}^h$ is the attention weights, $\Delta\boldsymbol{c}_{ij}$ is the delta coordinate between $\boldsymbol{c}_i$ and $\boldsymbol{c}_j$ where the superscript 0, 1 and 2 represent the $X$ axis, $Y$ axis and $Z$ axis respectively. Then the position prediction head predicts coordinate updates using three linear projections of the attention head values onto the three axes, which is denoted as:

$$\boldsymbol{o}^0 = \text{Concat}_h(\boldsymbol{A}^{(h,0)}\boldsymbol{V}^h); \boldsymbol{o}^0 = \text{Linear}_1(\boldsymbol{o}^0);$$
$$\boldsymbol{o}^1 = \text{Concat}_h(\boldsymbol{A}^{(h,1)}\boldsymbol{V}^h); \boldsymbol{o}^1 = \text{Linear}_2(\boldsymbol{o}^1);$$
$$\boldsymbol{o}^2 = \text{Concat}_h(\boldsymbol{A}^{(h,2)}\boldsymbol{V}^h); \boldsymbol{o}^2 = \text{Linear}_3(\boldsymbol{o}^2); \tag{7}$$
$$\Delta\boldsymbol{c}' = \text{Concat}([\boldsymbol{o}^0, \boldsymbol{o}^1, \boldsymbol{o}^2]); \boldsymbol{c}' = \boldsymbol{c} + \Delta\boldsymbol{c}';$$

where $\Delta\boldsymbol{c}'$ is the predicted coordinate updates and $\boldsymbol{c}'$ is the predicted coordinates.

As described in the above formula, the coordinate prediction head used in our study does not inherently enforce strict equivariance. This challenge can be addressed through one of two strategies: (1) Strict equivariance of the model can be achieved by sharing the parameters across the three linear layers in Eq. (7)-denoted as `linear_1`, `linear_2`, and `linear_3`-and concurrently eliminating the bias terms within these layers; (2) the model's robustness to spatial transformations can be enhanced by incorporating random rotations into the input coordinates as a form of data augmentation. During our experimental phase, both techniques were rigorously tested. The latter approach-data augmentation via random rotations yielded better accuracy in quantum chemistry property predictions and was thus selected for our model architecture. In this case, empirical evidence suggests that with a sufficiently large training dataset, such as the PCQM4MV2 dataset, the model naturally tends towards an equivariant state. Specifically, our observations indicate that the parameters of the three linear layers tend to converge to the same, and the bias terms asymptotically approach zero, with the discrepancies being marginal (on the order of $1e-4$).

## Training strategy

In DFT conformation optimization or Molecular Dynamics simulations, a conformation is optimized step-by-step, resulting in a trajectory from a raw conformation to the equilibrium conformation in Euclidean space. However, saving such a trajectory can be expensive, and publicly available datasets usually provide the equilibrium conformations only. Providing a trajectory would be beneficial as intermediate states can be used as data augmentation to guide the model's training. Inspired by this, we propose a novel training approach, which generates a pseudo trajectory first, samples a conformation from it, and uses the sampled conformation as input to predict the equilibrium conformation. This approach allows us to better exploit the information in the molecular data, which we found can greatly improve the model's performance. Specifically, we assume that the trajectory from a raw conformation $r^{init}$ to a target equilibrium conformation $r^{tgt}$ is a linear process. We generate an intermediate conformation along this trajectory via noisy interpolation, i.e.,

$$r^{(0)} = qr^{init} + (1-q)(r^{tgt} + c),\qquad(8)$$

where scalar $q$ ranges from 0 to 1, the Gaussian noise $c \in \mathbb{R}^{n \times 3}$ has a mean of 0 and standard deviation $v$ (a hyper-parameter). Taking $r^{(0)}$ as input, `Uni-Mol+` learns to update towards the target equilibrium conformation $r^{tgt}$. During inference, $q$ is set to 1.0 by default. However, during training, simply sampling $q$ from a uniform distribution ([0.0, 1.0]) may cause (1) a distributional shift between training and inference, due to the infrequent sampling of $q=1.0$ (RDKit-generated conformation), and (2) an inability to learn an accurate mapping from the equilibrium conformation to the QC properties, as $q=0.0$ (target conformation) is also not sampled often. Therefore, we employ a mixture of Bernoulli and Uniform distributions to flexibly assign higher sample probabilities to $q=1.0$ and $q=0.0$, while also sampling from interpolations. The above process is illustrated in Fig. 1c in Supplementary.

The model takes $r^{(0)}$ as input and generates $r^{(R)}$ after $R$ iterations. Then, the model uses $r^{(R)}$ as input and predicts the QC properties. L1 loss is applied to the QC property regression and the 3D coordinate prediction. All loss calculations are performed solely on the final conformer at the last iteration.

## Model configuration

Similar to both Graphormer[4] and Transformer-M[3], `Uni-Mol+` comprises 12 layers with an atom representation dimension of $d_x = 768$ and a pair representation dimension of $d_p = 256$. The hidden dimension of `FFN` in the atom representation track is set to 768, while that of the pair representation track is set to 256. Additionally, the hidden dimension in the `OuterProduct` is $d_o = 32$, and the hidden dimension in the `TriangularUpdate` is $d_t = 32$ as well. The number of conformation optimization iterations $R$ is set to 1, indicating that the model iterates twice in total (once for conformation optimization and once for quantum chemistry property prediction). For the training strategy, we specified a standard deviation of $v = 0.2$ for random noise and employed a particular sampling method for $q$. Specifically, $q$ was set to 0.0 with probability 0.8, set to 1.0 with probability 0.1, and uniformly sampled from [0.4, 0.6] with probability 0.1. With this setting, the number of parameters of `Uni-Mol+` is about 52.4M.

### Setting for PCQM4MV2

We used the AdamW optimizer with a learning rate of $2e-4$, a batch size of 1024, $(\beta_1, \beta_2)$ set to (0.9, 0.999), and gradient clipping set to 5.0 during training, which lasted for 1.5 million steps, with 150K warmup steps. Additionally, an exponential moving average (EMA) with a decay rate of 0.999 was utilized. The training took approximately 5 days, utilizing 8 NVIDIA A100 GPUs. The inference on the 147k test-dev set took approximately 7 minutes, utilizing 8 NVIDIA V100 GPUs.

### Setting for OC20

We use the default 12-layer `Uni-Mol+` setting for OC20 experiments. The model configuration deviates slightly from the settings employed in PCQM4MV2. Firstly, since OC20 lacks graph information, graph-related features are excluded from the model. Secondly, due to the greater number of atoms present in OC20 compared to PCQM4MV2, the model capacity is marginally reduced for efficiency reasons. In particular, the pair representation dimension $d_p$ is set to 128, while the hidden dimensions in the `OuterProduct` and `TriangularUpdate` are set to $d_o = 16$ and $d_t = 16$, respectively. Third, the periodic boundary condition needs to be considered; we adopt the solution proposed in[5], which pre-expands the neighbor cells and then applies a radius cutoff to reduce the number of atoms. The AdamW optimizer was employed during the training process, which lasted for 1.5 million steps, including 150K warmup steps. The optimizer was configured with a learning rate of $2e-4$, a batch size of 64, $(\beta_1, \beta_2)$ values of (0.9, 0.999), and a gradient clipping parameter of 5.0. The training process spanned approximately 7 days and made use of 16 NVIDIA A100 GPUs.

## Data availability

The datasets used in this study are all publicly available. PCQM4MV2 dataset which is to predict HOMO-LUMO GAP on small molecules is available at https://ogb.stanford.edu/docs/lsc/pcqm4mv2/#dataset and the OC20 dataset which is to conduct energy prediction on catalyst system is available at https://github.com/Open-Catalyst-Project/ocp/blob/main/DATASET.md. Source data are provided with this paper.

## Code availability

The source code of this study is publicly available on GitHub(https://github.com/deepmodeling/Uni-Mol/) and zenodo (https://doi.org/10.5281/zenodo.12670462) to allow replication of the results.

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

## Acknowledgements

We thank Bohang Zhang, Siyuan Liu and Hang Zheng for their helpful suggestion and discussion. Di He was supported by National Key R&D Program of China (2022ZD0160300) and National Science Foundation of China (NSFC62376007).

## Author contributions

S.L. and G.K. designed the model, conducted the experiments, and wrote the paper. Z.G. and D.H. assisted in writing and designing experiments. G.K. and L.Z. secured funding. All have commented on and edited the manuscript.

## Competing interests

The authors declare no competing interests.
