## [Peer Review File · Nature Communications]

REVIEWER COMMENTS

Reviewer #1 (Remarks to the Author):

Highly Accurate Quantum Chemical Property Prediction with Uni-Mol+

Key results:

This manuscript proposes to predict quantum properties of molecules, in specific the energy or homo-lumo gap calculated in DFT methods, via iteratively refining the RDKit-generated initial conformation. Experimental results on PCQM4M(v2) and OC20 benchmarks demonstrate the state-of-the-art (SOTA) performance of Uni-Mol+.

Merits:

The experimental results of Uni-Mol+ is comprehensive with proper ablation studies.

The proposed Uni-Mol+ achieved the SOTA performance on both PCQM4M(v2) and OC20 property prediction tasks.

Better contextualize the proposed model:

Since the backbone model of Uni-Mol+ has heavily overlapped with the Evoformer of AlphaFold2 (AF2) [1], it is better to cite AF2 properly in the main text (Section 2 & 4) to provide appropriate context for common readers. However, I only found the reference of AF2 in the supplementary material in the present version.

How do the authors conduct the model architecture design and compare it to AF2/Evoformer? To be specific, given the current architecture is based on AF2/Evoformer, what are the differences between these two? If there exists any architecture deviation, what is the rationale for such model design?

Originality and significance:

As the authors put, "our work emphasizes a novel paradigm for QC property prediction, rather than developing a new model backbone." How do the authors compare the Uni-Mol+ with the Uni-Mol [2] (3D position recovery) + Noisy Node [3] (interpolation strategy)? Though the original Uni-Mol pretraining does not involve energy prediction as a self-supervised task, one can easily append a property prediction

head on top of its architecture to do supervised pre-training. Could the author offer more insights beyond such a combination?

Generally speaking, the proposed method itself reveals few insights to the research community and thus lacks enough significance. The task is well-studied while the model architecture / training objectives lack novelty.

Suggested Complementary Experiments and Ablations:

Since the Uni-Mol+ requires RDKit to generate the initial 3D conformation as input which is different from previous baselines, could the author provide an ablation study demonstrating the significance of such a module? To be specific, how is the performance of Uni-Mol+ when the starting 3D conformation is noisy or not as good as RDKit? Is the model showing consistency or robustness for the evaluation results by switching different conformer sampling modules, including but not limited to the Openbabel mentioned in the paper?

Could the authors provide the benchmark results on the QM9 to make a fair comparison with the baseline models such as Transformer-M?

By comparing the Uni-Mol and Uni-Mol+ (this paper), the Uni-Mol involved the 3D position predictions task as the pre-training objectives while Uni-Mol+ merged it and the property prediction task into one single supervised training. Since one of the biggest merits of pre-training is to amortize the expensive training overhead and make finetuning accessible on downstream tasks, I am curious: what is the estimated training time for Uni-Mol+ for each task?

Could the author report the error bar or standard deviation for the evaluation metrics such as MAE?

How is the weight q encoded in the model? It would be good to see how the prediction performance changes along the pseudo trajectory with different q . The author can simply do an experiment by feeding the model with different noise interpolation q during inference.

Clarity of the Implementation details:

One of the core modules used in Uni-Mol+ is the RDKit which generates the initial 3D conformation of each input molecule. However, since RDKit is just a molecular toolkit, could the author please provide how they conduct such a generation pipeline in detail? For example, common practice using RDKit for conformer sampling involves the ETKDG [4] method to embed conformers in 3D space and apply MMFF94 for optimization. Is this the case for the Uni-Mol+? Plus, do the authors perform proper filtering for the RDKit-generated conformations? Please elaborate on this point.

One important component of the proposed Uni-Mol+ is the 3D position prediction head which enhances the supervision signals during training. However, there is no apparent description of the position prediction head can be found in the manuscript. Can the authors elaborate on this? Moreover, since the position is the molecular geometry in 3D space, how is the equivariance w.r.t. the spatial transformation considered in the architecture?

References:

- [1] Jumper, J., Evans, R., Pritzel, A., Green, T., Figurnov, M., Ronneberger, O., Tunyasuvunakool, K., Bates, R., Žídek, A., Potapenko, A. and Bridgland, A., 2021. Highly accurate protein structure prediction with AlphaFold. *Nature*, 596(7873), pp.583-589.
- [2] Zhou, G., Gao, Z., Ding, Q., Zheng, H., Xu, H., Wei, Z., Zhang, L. and Ke, G., 2023. Uni-Mol: a universal 3D molecular representation learning framework.
- [3] Godwin, J., Schaarschmidt, M., Gaunt, A., Sanchez-Gonzalez, A., Rubanova, Y., Veličković, P., Kirkpatrick, J. and Battaglia, P., 2021. Simple gnn regularisation for 3d molecular property prediction & beyond. arXiv preprint arXiv:2106.07971.
- [4] Riniker, S. and Landrum, G.A., 2015. Better informed distance geometry: using what we know to improve conformation generation. *Journal of chemical information and modeling*, 55(12), pp.2562-2574.

*Related links (just sketchy note):

Uni-Mol: a universal 3D molecular representation learning framework

Simple gnn regularisation for 3d molecular property prediction & beyond

One transformer can understand both 2d & 3d molecular data

Gps++: An optimised hybrid mpnn/transformer for molecular property prediction

Guideline for NC peer review:

Writing the review

The primary purpose of the review is to provide the editors with the information needed to reach a decision but the review should also instruct the authors on how they can strengthen their paper to the point where it may be acceptable. As far as possible, a negative review should explain to the authors the major weaknesses of their manuscript, so that rejected authors can understand the basis for the decision and see in broad terms what needs to be done to improve the manuscript for publication elsewhere. Referees should be aware that when declined manuscripts are transferred to another journal in the Nature Portfolio portfolio the referee comments are also transferred, and can be used to determine suitability of publication at the receiving journal. In the case of manuscript transfers between Nature Portfolio journals with in-house editors, referee identities are also transferred.

Confidential comments to the editor are welcome, but they should not contradict the main points as stated in the comments for transmission to the authors.

We ask reviewers the following questions, to provide an assessment of the various aspects of a manuscript:

Key results: Please summarise what you consider to be the outstanding features of the work.

Validity: Does the manuscript have flaws which should prohibit its publication? If so, please provide details.

Originality and significance: If the conclusions are not original, please provide relevant references. On a more subjective note, do you feel that the results presented are of immediate interest to many people in your own discipline, and/or to people from several disciplines?

Data & methodology: Please comment on the validity of the approach, quality of the data and quality of presentation. Please note that we expect our reviewers to review all data, including any extended data and supplementary information. Is the reporting of data and methodology sufficiently detailed and transparent to enable reproducing the results?

Appropriate use of statistics and treatment of uncertainties: All error bars should be defined in the corresponding figure legends; please comment if that's not the case. Please include in your report a

specific comment on the appropriateness of any statistical tests, and the accuracy of the description of any error bars and probability values.

Conclusions: Do you find that the conclusions and data interpretation are robust, valid and reliable?

Suggested improvements: Please list additional experiments or data that could help strengthening the work in a revision.

References: Does this manuscript reference previous literature appropriately? If not, what references should be included or excluded?

Clarity and context: Is the abstract clear, accessible? Are abstract, introduction and conclusions appropriate?

Inflammatory material: Does the manuscript contain any language that is inappropriate or potentially libelous?

Springer Nature is committed to diversity, equity and inclusion; please raise any concerns that may in your view have an impact on this commitment.

Please indicate any particular part of the manuscript, data, or analyses that you feel is outside the scope of your expertise, or that you were unable to assess fully.

Please address any other specific question asked by the editor via email.

Reports do not necessarily need to follow this specific order but should document the referees' thought process. All statements should be justified and argued in detail, naming facts and citing supporting references, commenting on all aspects that are relevant to the manuscript and that the referees feel qualified commenting on. Not all of the above aspects will necessarily apply to every paper, due to discipline-specific standards. When in doubt about discipline-specific refereeing standards, reviewer can contact the editor for guidance.

It is our policy to remain neutral with respect to jurisdictional claims in published maps and institutional affiliations, and the naming conventions used in maps and affiliation are left to the discretion of authors. Referees should not, therefore, request authors to make any changes to such unless it is critical to the clarity of the scientific content of a manuscript

Reviewer #2 (Remarks to the Author):

The authors study a significant problem predicting quantum properties from 2D molecule modality, which usually requires time-consuming DFT computation. Specifically, the author propose the Uni-Mol+ framework that models the precise molecule geometry from coarse ones given by RDKit. Experimental

results and ablations show a significant performance improvement on multiple benchmarks and indicate the effectiveness of the proposed approach.

Below are some itemized concerns and suggestions:

1. The authors provide parameter comparison on multiple datasets, but the time cost is missing. In addition to the parameters, the iterations R is another variable to determine the overall computation cost. Since the goal of the studied problem is, in essence, to approximating DFT computation with affordable computation, it would be important to compare practical time cost among different approaches as well as DFT. In addition, could the authors perform study the effect of increasing R on molecules grouped by different sizes? Given the intuition that DFT computation cost grows significantly over molecule size, it's reasonable to assume that more computation is required for larger molecules, controlled by R in Uni-Mol+. Therefore it's important to make clear the scalability of Uni-Mol+.
2. The propose training strategy is novel and effective. And I appreciate the authors' discussion on the difference between the proposed noisy interpolation with Noisy Node. I wonder if the authors can further perform ablations comparing the two approach. It will add to the significance of this work since the noisy interpolation may be used as a fundamental technique in different task, modalities, and models.
3. The authors claim that their method refines initial conformations towards DFT equilibrium conformation. I suggest a quantitative evaluation of the optimized conformation's accuracy. For instance, comparing the RMSD between the refined conformation and the ground truth would be informative (beyond the selected samples' RMSD presented in Figure 2). Additionally, evaluating the performance of trained 3D GNN models, like SphereNet, in predicting QC properties from the optimized molecular geometries would be beneficial, considering the known accuracy of 3D GNN models in predicting QC properties from equilibrium conformations.
4. In the OGB leaderboard, a method named "EGT+Tri. Attn.+RDKit Coords." outperforms Uni-Mol+. I recommend that the authors include this method in Table 1 for a comprehensive comparison.
5. Figure 3 currently only shows the energy difference between initial and predicted conformers, which offers limited information. Providing the energy difference between initial and equilibrium conformers would more accurately reflect the model's performance in optimizing molecular geometries.

6. Could the authors provide further insights into why integrating explicit 3D geometry prediction within the neural network is superior to maximizing mutual information between 2D and 3D molecular views during training?

Below are some additional questions:

1. The use of RDKit for generating initial conformations introduces some randomness. How does this affect the accuracy of QC property predictions? Additionally, how robust is the proposed method in optimizing equilibrium conformations from various initial conformations?

2. When the iteration count (R) is 1, leading to the prediction of two conformers, is the L1 loss on structures calculated for each predicted conformer or only the final one? Furthermore, is the QC property loss assessed based solely on the final conformer?

Response to Reviewers' Comments on "Highly Accurate Quantum Chemical Property Prediction with Uni-Mol+"

Response to Reviewer #1

We are grateful to the reviewer for the insightful feedback. Here is our detailed response to each of
your comments.

1.1 Comparison with Evoformer of AlphaFold2

▷ Review Comment: *Better contextualize the proposed model: Since the backbone model of Uni-Mol+ has*
*heavily overlapped with the Evoformer of AlphaFold2 (AF2) [1], it is better to cite AF2 properly in the main text*
*(Section 2 & 4) to provide appropriate context for common readers. However, I only found the reference of AF2*
*in the supplementary material in the present version. How do the authors conduct the model architecture design*
*and compare it to AF2/Evoformer? To be specific, given the current architecture is based on AF2/Evoformer,*
*what are the differences between these two? If there exists any architecture deviation, what is the rationale for*
*such model design?*

Thank you for your suggestion. We've updated our manuscript to include references to AlphaFold2
in Sections 2 and 4.

We agree that at a concept level, Uni-Mol+ shares similarities with AlphaFold2's Evoformer, espe-
cially in how both use a two-track transformer approach (Node Representation and Pair Representa-
tion). However, the key difference lies in the type of input data each one uses. Evoformer is built to
process protein sequences and multiple sequence alignments (MSAs), whereas Uni-Mol+ is designed
for atoms and their 3D positions. This fundamental difference leads to several variations in their
design.

For instance, Evoformer includes specific layers like MSARowAttention and MSAColumnAttention
to handle MSAs, which are not needed in Uni-Mol+. In the Pair Representation, Uni-Mol+ uses
a single TriangularUpdate for efficiency, while Evoformer uses four different Triangular operators.
Furthermore, Evoformer relies on a complex StructureModule to predict the protein structure from
sequence representation, while Uni-Mol+ simply needs a SE(3) coordinate head. As a result, Uni-
Mol+ is simpler and more efficient compared to Evoformer. We have elaborated on these differences
in Supplementary Section 1.

1.2 Comparison with Uni-Mol and Noisy Nodes

▷ Review Comment: *As the authors put, "our work emphasizes a novel paradigm for QC property prediction,*
*rather than developing a new model backbone." How do the authors compare the Uni-Mol+ with the Uni-Mol*
*[2] (3D position recovery) + Noisy Node [3] (interpolation strategy)? Though the original Uni-Mol pretraining*
*does not involve energy prediction as a self-supervised task, one can easily append a property prediction head*
*on top of its architecture to do supervised pre-training. Could the author offer more insights beyond such a*
*combination? Generally speaking, the proposed method itself reveals few insights to the research community*
*and thus lacks enough significance. The task is well-studied while the model architecture / training objectives*
*lack novelty.*

We’ve included two additional ablation study results in our paper for a more comprehensive
comparison. These can be found as entries No.18 and No.19 in the below Table 1 (and Table 3 in
the revised manuscript).

Firstly, when we compare the results of No.18 and No.1, it’s clear that the performance of Noisy
Nodes (No.18, result 0.0760) is significantly lower than that of Uni-Mol+ (No.1, result 0.0696). It’s
important to note that the only difference between these two is the training strategy, while the model
structure remains the same. This large performance gap (0.0760 vs. 0.0696) highlights the importance
of our proposed training strategy. As we explained in Section 3, our training approach is effective
because it additionally uses a Bernoulli distribution, which helps in dealing with distributional shifts
and improves predictions of quantum chemistry (QC) properties.

Secondly, a comparison between No.19 and No.18 shows that using the Uni-Mol’s backbone yields
slightly worse results than using Uni-Mol+’s backbone. However, this difference is not very large.

These additional ablation studies confirm that Uni-Mol+ largely outperforms the combination of
Uni-Mol and Noisy Nodes. The key factor behind this improvement is our proposed training strategy.
This supports our claim that “our work emphasizes a novel paradigm for QC property prediction,
rather than developing a new model backbone”. And we have elaborated more discussions about this
in Sec. 2.4.

Table 1: Comparison with Noisy Nodes and Uni-Mol.

No.	Model Backbone	Training Strategy	Valid MAE (\downarrow)
1	Uni-Mol+	Uni-Mol+	0.0696
...			
18	Uni-Mol+	Noisy Nodes	0.0760
19	Uni-Mol	Noisy Nodes	0.0798

1.3 More Experiments

1.3.1 Robustness regarding initial conformations

\triangleright Review Comment: *Since the Uni-Mol+ requires RDKit to generate the initial 3D conformation as input which*
*is different from previous baselines, could the author provide an ablation study demonstrating the significance of*
*such a module? To be specific, how is the performance of Uni-Mol+ when the starting 3D conformation is noisy*
*or not as good as RDKit? Is the model showing consistency or robustness for the evaluation results by switching*
*different conformer sampling modules, including but not limited to the Openbabel mentioned in the paper?*

Thank you for the valuable suggestion. We have conducted additional experiments to assess the
robustness of our model with varying input conformations. Specifically, we introduced Gaussian
noise (with standard deviations of 0.1 and 0.3) to the initial RDKit conformations. The results, as
detailed in Table 2, demonstrate that our model’s performance is relatively unaffected by changes in
the initial conformations.

Furthermore, we conducted an experiment starting from 2D conformations (with a flat z-axis) gen-
erated by RDKit’s `AllChem.Compute2DCoords`. Despite the significant challenge posed by the
absence of 3D information, the result is only a minor drop in performance and still largely outper-
form previous baselines. This finding underscores the robustness of Uni-Mol+: it maintains high
performance levels even without 3D conformation inputs. We have elaborated on these discussions in
Supplementary Section 3.

1.3.2 QM9 experiments

Table 2: The benchmark results on PCQM4MV2, with different initial conformations.

Method	Valid MAE (\downarrow)
Uni-Mol+	0.0695
Uni-Mol+ w/ Noisy RDKit Conf., std=0.1	0.0695
Uni-Mol+ w/ Noisy RDKit Conf., std=0.3	0.0694
Uni-Mol+ w/ 2D Conf.	0.0715

\triangleright Review Comment: *Could the authors provide the benchmark results on the QM9 to make a fair comparison*
 *with the baseline models such as Transformer-M?*

For QM9, the task involves directly learning quantum chemistry (QC) properties based on DFT
 equilibrium conformation. However, Uni-Mol+ is designed to focus on tasks that do not require
 knowledge of DFT conformation during inference. As a result, direct comparisons and results on the
 QM9 dataset may not be appropriate for evaluating Uni-Mol+’s performance.

1.3.3 Training costs and comparison with pretraining

\triangleright Review Comment: *By comparing the Uni-Mol and Uni-Mol+ (this paper), the Uni-Mol involved the 3D*
 *position predictions task as the pre-training objectives while Uni-Mol+ merged it and the property prediction*
 *task into one single supervised training. Since one of the biggest merits of pre-training is to amortize the*
 *expensive training overhead and make finetuning accessible on downstream tasks, I am curious: what is the*
 *estimated training time for Uni-Mol+ for each task?*

As noted in Supplementary Section 2, training for the PCQM4MV2 task required approximately
 5 days on 8 NVIDIA A100 GPUs, while the OC20 task took around 7 days on 16 NVIDIA A100
 GPUs.

We acknowledge the benefits of pretraining, particularly in improving data efficiency for downstream
 tasks and potentially reducing training costs. Pretraining is especially beneficial for enhancing
 performance in tasks with limited labeled data, such as ADME/T tasks in molecular representation
 learning. Yet, for data-rich tasks such as PCQM4MV2 and OC20, the immediate need for pretraining
 to enhance performance is reduced. Crucially, pretraining is not only compatible with Uni-Mol+ but
 also complementary. In future work, we believe that incorporating pretraining could further improve
 Uni-Mol+’s data efficiency.

1.3.4 Error bars

\triangleright Review Comment: *Could the author report the error bar or standard deviation for the evaluation metrics such*
 *as MAE?*

We appreciate your suggestion and have accordingly trained additional Uni-Mol+ models using two
 distinct seeds on both the PCQM4MV2 and OC20 benchmarks. We’ve computed the mean and
 standard deviation of the inference results and updated the experimental tables (Table 1 and Table 2
 in our manuscript). The minimal variation in the results confirms Uni-Mol+’s robust performance,
 consistently outperforming previous models despite accounting for randomness.

It’s important to note, however, that due to leaderboard submission constraints, we’re unable to
 provide error bars for the test sets. Moreover, previous baseline publications did not report error
 bars, and replicating their models to calculate these would be challenging. We are grateful for your
 insightful suggestions and have endeavored to provide the most comprehensive and reliable data
 within these constraints.

1.3.5 Strategy of sampling q

▷ Review Comment: *How is the weight q encoded in the model? It would be good to see how the prediction*
*performance changes along the pseudo trajectory with different q . The author can simply do an experiment by*
*feeding the model with different noise interpolation q during inference.*

As detailed in Section 4.2, during training, the sampling strategy of q uses a mixture of Bernoulli
distribution and Uniform distribution. The Bernoulli distribution addresses (1) the distributional
shift between training and inference and (2) enhances the learning of an accurate mapping from
the equilibrium conformation to the QC properties. Meanwhile, the Uniform distribution generates
additional intermediate states to serve as model inputs, effectively augmenting the input conformations.
During inference, since the DFT equilibrium conformations are unknown, q is fixed to 1.0.

We value your input on evaluating inference performance across pseudo trajectories with varying q
values. However, it’s crucial to mention that the available validation and test sets in the experimental
datasets **do not provide DFT equilibrium conformations**, rendering the proposed experiment
unfeasible with the current data. Nevertheless, given that quantum chemistry (QC) properties are
calculated from DFT equilibrium conformations, it’s reasonable to assume that smaller q values
(which are closer to equilibrium), would yield better results.

1.4 Clarity of the Implementation details

▷ Review Comment: *Clarity of the Implementation details: One of the core modules used in Uni-Mol+ is the*
*RDKit which generates the initial 3D conformation of each input molecule. However, since RDKit is just a*
*molecular toolkit, could the author please provide how they conduct such a generation pipeline in detail? For*
*example, common practice using RDKit for conformer sampling involves the ETKDG [4] method to embed*
*conformers in 3D space and apply MMFF94 for optimization. Is this the case for the Uni-Mol+? Plus, do*
*the authors perform proper filtering for the RDKit-generated conformations? Please elaborate on this point.*
*One important component of the proposed Uni-Mol+ is the 3D position prediction head which enhances the*
*supervision signals during training. However, there is no apparent description of the position prediction head*
*can be found in the manuscript. Can the authors elaborate on this? Moreover, since the position is the molecu-*
*lar geometry in 3D space, how is the equivariance w.r.t. the spatial transformation considered in the architecture?*

Thank you for your inquiry regarding the implementation details. In response to your question
about the use of RDKit for conformer sampling in our study, we indeed employ the ETKDG
method to generate 3D conformations, followed by MMFF94 force field for optimization. In
molecules where the generation of a 3D conformation is unsuccessful, we default to producing a
2D conformation with a flat z-axis using RDKit’s AllChem.Compute2DCoords function instead.
There are no additional filtering rules. Our preprocessing scripts are publicly available for re-
view and can be accessed at the following GitHub repository: [https://github.com/dptech-corp/Uni-](https://github.com/dptech-corp/Uni-Mol/blob/main/unimol_plus/scripts/get_3d_lmdb.py)
[Mol/blob/main/unimol_plus/scripts/get_3d_lmdb.py](https://github.com/dptech-corp/Uni-Mol/blob/main/unimol_plus/scripts/get_3d_lmdb.py).

Regarding the 3D position prediction head within Uni-Mol+, we have adopted the 3D prediction head
proposed in Graphormer-3D [1]. The architecture takes atom representation \mathbf{x}^L , pair representation
\mathbf{p}^L , and initial coordinates \mathbf{c} as inputs. An attention mechanism is initially employed and then the
attention weights is multiplied point-wisely with the pairwise delta coordinates derived from the
initial coordinates. The attention mechanism is denoted as:

$$\begin{aligned} \mathbf{Q}^h &= \mathbf{x}^L \mathbf{W}_Q^h; & \mathbf{K}^h &= \mathbf{x}^L \mathbf{W}_K^h; \\ \mathbf{B}^h &= \mathbf{p}^L \mathbf{W}_B^h; & \mathbf{V}^h &= \mathbf{x}^L \mathbf{W}_V^h; \\ \mathbf{A}_{i,j}^h &= \text{softmax} \left(\frac{\mathbf{Q}_i^h (\mathbf{K}_j^h)^T}{\sqrt{d_h}} + \mathbf{B}_{i,j}^h \right); \\ \Delta(\mathbf{c})_{i,j} &= \mathbf{c}_i - \mathbf{c}_j; & \mathbf{A}_{i,j}^{(h,0)} &= \mathbf{A}_{i,j}^h \odot \Delta(\mathbf{c})_{i,j}^0; \\ \mathbf{A}_{i,j}^{(h,1)} &= \mathbf{A}_{i,j}^h \odot \Delta(\mathbf{c})_{i,j}^1; & \mathbf{A}_{i,j}^{(h,2)} &= \mathbf{A}_{i,j}^h \odot \Delta(\mathbf{c})_{i,j}^2; \end{aligned} \quad (1)$$

where d_h is the head dimension, $\mathbf{W}_Q^h, \mathbf{W}_K^h, \mathbf{W}_V^h \in \mathbb{R}^{d_x \times d_h}$, $\mathbf{W}_B^h \in \mathbb{R}^{d_p \times 1}$ and $\mathbf{B}_{i,j}^h$ is an attention
 bias term. \mathbf{A}^h is the attention weights, $\Delta(\mathbf{c})_{ij}$ is the delta coordinate between \mathbf{c}_i and \mathbf{c}_j where
 the superscript 0, 1 and 2 represent the X axis, Y axis and Z axis respectively. Then the position
 prediction head predicts coordinate updates using three linear projections of the attention head values
 onto the three axes, which is denoted as:

$$\begin{aligned} \mathbf{o}^0 &= \text{Concat}_h(\mathbf{A}^{(h,0)} \mathbf{V}^h), & \mathbf{o}^0 &= \text{Linear}_1(\mathbf{o}^0); \\ \mathbf{o}^1 &= \text{Concat}_h(\mathbf{A}^{(h,1)} \mathbf{V}^h), & \mathbf{o}^1 &= \text{Linear}_2(\mathbf{o}^1); \\ \mathbf{o}^2 &= \text{Concat}_h(\mathbf{A}^{(h,2)} \mathbf{V}^h), & \mathbf{o}^2 &= \text{Linear}_3(\mathbf{o}^2); \\ \Delta(\mathbf{c}') &= \text{Concat}([\mathbf{o}^0, \mathbf{o}^1, \mathbf{o}^2]), & \mathbf{c}' &= \mathbf{c} + \Delta(\mathbf{c}'); \end{aligned} \tag{2}$$

where $\Delta(\mathbf{c}')$ is the predicted coordinate updates and \mathbf{c}' is the predicted coordinates.

Regarding the equivariance, as described in the above formula, the coordinate prediction head used in
 our study does not inherently enforce strict equivariance. This challenge can be addressed through
 one of two strategies: (1) Strict equivariance of the model can be achieved by sharing the parameters
 across the three linear layers in Equation (2)—denoted as `linear1`, `linear2`, and `linear3`—and
 concurrently eliminating the bias terms within these layers; (2) the model’s robustness to spatial
 transformations can be enhanced by incorporating random rotations into the input coordinates as
 a form of data augmentation. During our experimental phase, both techniques were rigorously
 tested. The latter approach—data augmentation via random rotations—yielded better accuracy in
 quantum chemistry property predictions and was thus selected for our model architecture. In this case,
 empirical evidence suggests that with a sufficiently large training dataset, such as the PCQM4MV2
 dataset, the model naturally tends towards an equivariant state. Specifically, our observations indicate
 that the parameters of the three linear layers tend to converge to the same, and the bias terms
 asymptotically approach zero, with the discrepancies being marginal (on the order of 1e-4).

We have elaborated on these details in the main text (Section 2.2) and Supplementary Section 1. We
 appreciate your request for clarification and hope this response adequately addresses your concerns.

[1] Yu Shi et al. Benchmarking graphormer on large-scale molecular modeling datasets, 2022.

2 Response to Reviewer #2

We greatly appreciate the reviewer’s constructive comments. Below, we provide a comprehensive
 response to each point raised.

2.1 Time Cost Regarding Molecular Sizes

*▷ Review Comment: 1. The authors provide parameter comparison on multiple datasets, but the time cost is*
 *missing. In addition to the parameters, the iterations R is another variable to determine the overall computation*
 *cost. Since the goal of the studied problem is, in essence, to approximating DFT computation with affordable*
 *computation, it would be important to compare practical time cost among different approaches as well as DFT.*
 *In addition, could the authors perform study the effect of increasing R on molecules grouped by different sizes?*
 *Given the intuition that DFT computation cost grows significantly over molecule size, it’s reasonable to assume*
 *that more computation is required for larger molecules, controlled by R in Uni-Mol+. Therefore it’s important to*
 *make clear the scalability of Uni-Mol+.*

We appreciate your valuable feedback. Addressing your concerns, we’ve conducted a detailed analysis
 of the computational time costs, particularly focusing on different molecular sizes. For this analysis,
 we chose a diverse set of molecules, grouping them by size, and calculated the average time required
 to process a single molecule in each category, including 50 molecules per size group.

We compared the time costs between Uni-Mol+ across different numbers of conformation update
 rounds (R) and traditional Density Functional Theory (DFT) calculations. Specifically, for Uni-Mol+,
 we assessed the computational time for molecules with up to 256 atoms. However, due to the
 substantial time demands of DFT calculations, we limited our DFT time cost analysis to molecules
 with a maximum of 50 atoms.

The computational evaluations for Uni-Mol+ were performed on a single NVIDIA V100 GPU, while
 the DFT calculations, including geometry optimization and quantum chemical energy computations,
 were conducted using Psi4 1.4.1 on 32 CPUs. The DFT calculations employed the B3LYP functional
 and 6-31G* basis set, aligning with the settings used in the PCQM4MV2 dataset.

We have presented these results in Fig. 1. The results clearly indicate that Uni-Mol+ not only provides
 a faster computational solution compared to DFT but also demonstrates superior scalability in relation
 to molecular size. Moreover, our findings reveal that the computational time cost associated with
 Uni-Mol+ increases linearly with the number of update rounds (R), affirming the predictability and
 efficiency of our method. We have elaborated on these discussions in Supplementary Section 3.

Figure 1: Time cost for different molecular sizes. (a) DFT versus Uni-Mol+ (with varying update rounds R); (b) Effect of different update rounds R in Uni-Mol+.

2.2 Comparison with Noisy Nodes

\triangleright Review Comment: 2. *The propose training strategy is novel and effective. And I appreciate the authors'*
 *discussion on the difference between the proposed noisy interpolation with Noisy Node. I wonder if the authors*
 *can further perform ablations comparing the two approach. It will add to the significance of this work since the*
 *noisy interpolation may be used as a fundamental technique in different task, modalities, and models.*

We've included an additional ablation study result in our paper for a more comprehensive comparison,
 referring to No.18 in the below Table 3 (and Table 3 in the revised manuscript).

Compared the results of No.18 and No.1, it's clear that the performance of Noisy Nodes (No.18,
 result 0.0760) is significantly lower than that of Uni-Mol+ (No.1, result 0.0696). It's important to
 note that the only difference between these two is the training strategy, while the model structure
 remains the same. This large performance gap (0.0760 vs. 0.0696) highlights the importance of our
 proposed training strategy. As we explained in Section 3, our training approach is effective because
 it additionally uses a Bernoulli distribution, which helps in dealing with distributional shifts and
 improves predictions of quantum chemistry (QC) properties.

Table 3: Comparison with Noisy Nodes.

No.	Model Backbone	Training Strategy	Valid MAE (\downarrow)
1	Uni-Mol+	Uni-Mol+	0.0696
...			
18	Uni-Mol+	Noisy Nodes	0.0760

2.3 Quantitative Analysis for the Optimized Conformations

▷ Review Comment: 3. The authors claim that their method refines initial conformations towards DFT equilibrium conformation. I suggest a quantitative evaluation of the optimized conformation’s accuracy. For instance, comparing the RMSD between the refined conformation and the ground truth would be informative (beyond the selected samples’ RMSD presented in Figure 2). Additionally, evaluating the performance of trained 3D GNN models, like SphereNet, in predicting QC properties from the optimized molecular geometries would be beneficial, considering the known accuracy of 3D GNN models in predicting QC properties from equilibrium conformations.

Thank you for your suggestion regarding the quantitative evaluation of our method’s accuracy in refining initial conformations towards DFT equilibrium conformation. We acknowledge the importance of a precise quantitative assessment in validating the efficacy of our approach.

It’s crucial to note that within the PCQM4MV2 benchmark, ‘ground truth’ conformations for validation or test sets are not readily available. Consequently, as explicitly mentioned in our manuscript, we resorted to generating DFT conformations independently. It is, however, imperative to recognize that these self-generated equilibrium conformations may not precisely mirror the authentic ground truth conformations intrinsic to the PCQM4MV2 dataset. This discrepancy inherently complicates the process of conducting an accurate quantitative evaluation of the optimized conformation’s precision. As a result, we initially limited our analysis to presenting a qualitative evaluation through selected examples.

Nonetheless, acknowledging the value of your suggestion, we ventured to extend our analysis to the OC20 benchmark, where ground-truth equilibrium conformations are accessible within the validation dataset. We have conducted additional evaluations and benchmarked our results against the current state-of-the-art model, EquiFormer. The results, as depicted in Table 4, clearly demonstrate that Uni-Mol+ surpasses the previous baseline in predicting equilibrium conformations. We have elaborated on these discussions in Supplementary Section 3.

Table 4: RMSD for predicted conformations on OC20 valid set.

Model	RMSD↓			
	ID	OOD Ads.	OOD Cat.	OOD Both
EquiFormer	1.7622	1.7277	1.0157	1.8116
Uni-Mol+	1.5067	1.4756	0.9281	1.5401

Regarding your suggestion about employing trained 3D GNN models, like SphereNet, to assess performance based on the predicted equilibrium conformations, we believe this step might not be necessary for two main reasons. Firstly, Uni-Mol+ can predict QC properties based on the optimized conformations without the need for intermediary models. Secondly, using DFT calculations to derive QC properties from the optimized conformations ensures greater accuracy than 3D GNN model-based predictions. This has been supported in our manuscript, specifically in Figure 3. The results presented in Figure 3 indicate that the conformations optimized by Uni-Mol+ reach a lower energy state compared to their initial states, underscoring the proficiency of Uni-Mol+ in conformation optimization.

2.4 Recent Leaderboard Results

▷ Review Comment: 4. In the OGB leaderboard, a method named “EGT+Tri. Attn.+RDKit Coords.” outperforms Uni-Mol+. I recommend that the authors include this method in Table 1 for a comprehensive comparison.

We sincerely appreciate the reviewer’s suggestion to include the “EGT+Tri. Attn.+RDKit Coords.” method in Table 1 for a comprehensive comparison. It’s important to note that “EGT+Tri. Attn.+RDKit Coords.” was first released in November 2023, approximately 8 months after the initial release of Uni-Mol+ in March 2023. Besides, “EGT+Tri. Attn.+RDKit Coords.” was significantly

influenced by the pioneering Uni-Mol+, incorporating approaches such as RDKit-generated conformations
and triangular operators, and also references Uni-Mol+’s paper in its publication. Furthermore,
at the time of submitting this manuscript in October 2023, “EGT+Tri. Attn.+RDKit Coords.” had
not yet featured in the OGB leaderboard. Given these considerations, directly comparing “EGT+Tri.
Attn.+RDKit Coords.” with Uni-Mol+ in Table 1 might not accurately represent the state of research
at the time of Uni-Mol+’s submission. To ensure clarity and provide context regarding the timeline of
the works cited, we have added a footnote in Table 1. This footnote explicitly mentions the dates
when the leaderboard was accessed, thereby maintaining the relevance and fairness of the comparison
within the evolving research environment.

2.5 Energy Difference Between Initial and Equilibrium Conformers

\triangleright Review Comment: 5. Figure 3 currently only shows the energy difference between initial and predicted
conformers, which offers limited information. Providing the energy difference between initial and equilibrium
conformers would more accurately reflect the model’s performance in optimizing molecular geometries.

We appreciate the reviewer’s suggestion to illustrate the energy difference between the initial and equi-
librium conformers, offering a more comprehensive view of the model’s performance in optimizing
molecular geometries.

However, it’s important to note that the PCQM4MV2 dataset’s validation set does not provide a
ground truth equilibrium conformation or its corresponding energy, which poses a challenge in
presenting these results directly. In response to this, we have independently calculated the equilibrium
conformation and its energy by DFT, and we present this additional information in the newly added
Fig. 2. This figure illustrates that the energy difference distribution between the initial and predicted
conformations closely aligns with that between the initial and equilibrium conformations. This
similarity demonstrates Uni-Mol+’s effectiveness in predicting equilibrium conformations accurately.
We have revised Figure 3 in our main text and elaborated more discussions in Section 3 in the main
text.

Figure 2: Distribution of delta energy. We selected 100 data points and used DFT to calculate the following values: (a) the delta energies between their initial and Uni-Mol+’s predicted conformations; (b) the delta energies between their initial conformations and the DFT conformations, where the DFT conformations are calculated by ourselves using DFT tool. Cross-marks indicate data points with increased energies, while circle-marks denote those with decreased energies.

2.6 Comparison with Mutual Information Between 2D and 3D Molecular Views

\triangleright Review Comment: 6. Could the authors provide further insights into why integrating explicit 3D geometry
prediction within the neural network is superior to maximizing mutual information between 2D and 3D

*molecular views during training?*

We value the inquiry into the advantages of integrating explicit 3D geometry prediction within our
neural network, as compared to methods that maximize mutual information between 2D and 3D
molecular views during training.

Quantum chemical (QC) properties are intrinsically calculated based on 3D equilibrium conforma-
tions. Hence, the most effective approach to predict these properties is to learn a direct mapping
from the 3D conformation to QC properties. However, given the challenges in obtaining equilibrium
conformations during inference, previous models primarily learn a mapping from 2D molecular
graphs to QC properties, denoted as $x_{2D} \rightarrow y$, where x_{2D} represents the 2D molecular graph input,
and y denotes a QC property. While some models aim to maximize mutual information between 2D
and 3D molecular views, represented as $x_{2D} \rightarrow (x_{3D}, y)$, this approach doesn’t explicitly learn a
mapping from the 3D equilibrium conformation x_{3D} to y . This is a crucial shortcoming since y is
highly correlated with x_{3D} .

Some models, like Transformer-M, attempt to learn both $x_{2D} \rightarrow y$ and $x_{3D} \rightarrow y$. However, during
inference, these models rely solely on x_{2D} , which compromises the prediction performance.

Uni-Mol+, on the other hand, employs a strategy $x'_{3D} \rightarrow \dots \rightarrow x_{3D} \rightarrow y$. This process starts with
a raw 3D conformation x'_{3D} , iteratively refines it towards x_{3D} , and then predicts y . By explicitly
learning a mapping from 3D conformation to QC properties, Uni-Mol+ proves to be more effective
than previous models. We have provided a detailed discussion on this matter in Section 3 of our
manuscript, clarifying Uni-Mol+’s superior performance and methodological advantages.

2.7 Robustness Regarding Initial Conformations

\triangleright Review Comment: *The use of RDKit for generating initial conformations introduces some randomness. How*
*does this affect the accuracy of QC property predictions? Additionally, how robust is the proposed method in*
*optimizing equilibrium conformations from various initial conformations?*

Thank you for the valuable suggestion. We have conducted additional experiments to assess the
robustness of our model with varying input conformations. Specifically, we introduced Gaussian
noise (with standard deviations of 0.1 and 0.3) to the initial RDKit conformations. The results, as
detailed in Table 5, demonstrate that our model’s performance is relatively unaffected by changes in
the initial conformations.

Furthermore, we conducted an experiment starting from 2D conformations (with a flat z-axis) gen-
erated by RDKit’s `AllChem.Compute2DCoords`. Despite the significant challenge posed by the
absence of 3D information, the result is only a minor drop in performance and still largely outper-
form previous baselines. This finding underscores the robustness of Uni-Mol+: it maintains high
performance levels even without 3D conformation inputs. We have elaborated on these discussions in
Supplementary Section 2.

Table 5: The benchmark results on PCQM4MV2, with different initial conformations.

Method	Valid MAE (\downarrow)
Uni-Mol+	0.0695
Uni-Mol+ w/ Noisy RDKit Conf., std=0.1	0.0695
Uni-Mol+ w/ Noisy RDKit Conf., std=0.3	0.0694
Uni-Mol+ w/ 2D Conf.	0.0715

2.8 Loss Calculation in Iterative Updates

\triangleright Review Comment: *When the iteration count (R) is 1, leading to the prediction of two conformers, is the L1 loss*
*on structures calculated for each predicted conformer or only the final one? Furthermore, is the QC property*

*loss assessed based solely on the final conformer?*

All loss calculations, including the L1 loss on structures and the QC property loss, are performed
solely on the final conformer at the last iteration. We have revised Section 4.2 to make this more
clear.

REVIEWER COMMENTS

Reviewer #1 (Remarks to the Author):

The authors addressed most of my comments. There are some further comments and questions below:

1.1 The mention is supposed to be "AlphaFold2" instead of "AlphaFold" in the revised manuscript. The author could revise correspondingly.

1.3.1 Since most of the conformation variety of molecules comes from rotating the covalent bond (i.e. the torsion angles), could the author provide results by randomly sampling alternative input structure via perturbing the torsion angle. You can use the conformation sampling model like [1]. Could the authors provide further results showing the behavior and robustness of Uni-Mol+ when inputting alternative conformations based on the torsion angle perturbations?

1.3.3 Thank the authors for clarifying. Could the author further discuss the following situation of Uni-Mol+: the SotA performance of Uni-Mol+ might rely on the fact that the involved two benchmark sets, namely PCQM4MV2 and OC20, includes large-scale DFT equilibrium conformations as well as labels for training. This is however not common to other real-world settings for molecular property prediction (PP) tasks, where the data can be insufficient. On the other hand, the "pre-training" based models aim to induce fast and effective domain adaptation via fine-tuning. In this sense, how do the authors see the potential applicability of this Uni-Mol+ framework when (3D conf, label) data can be short, or provide evidence to justify the performance of Uni-Mol+ in such a scenario?

[1] Jing, B., Corso, G., Chang, J., Barzilay, R., & Jaakkola, T. (2022). Torsional diffusion for molecular conformer generation. *Advances in Neural Information Processing Systems*, 35, 24240-24253.

Reviewer #2 (Remarks to the Author):

Thank you for the additional results and the comprehensive analysis. My concerns are mostly addressed in the revised manuscript except for one remaining request regarding "the effect of increasing R on

molecules grouped by different sizes". I appreciate the authors' study on the relationship between iterations "R" and computational cost (time). To study the effect, I suggest the authors also include the "error" dimension. For example, will larger molecules requires significantly more iterations to achieve a comparable error-level to DFT? This can be done by performing some per-size-group evaluations to see if, for larger molecules, there is a larger performance gap between Uni-Mol+ (with default $R \leq 2$) and DFT, and hence greater R is necessary. This could complete the scalability analysis and help the community to understand the limitation of deep learning-based approaches (if any).

Reviewer #2 (Remarks on code availability):

Code is runnable.

Response to Reviewers’ Comments on “Highly Accurate Quantum Chemical Property Prediction with Uni-Mol+”

Response to Reviewer #1

We are grateful to the reviewer for the additional feedback. Here is our detailed response to each of
your comments.

1.1 Revise "AlphaFold" to "AlphaFold2"

▷ Review Comment: *The mention is supposed to be "AlphaFold2" instead of "AlphaFold" in the revised*
*manuscript. The author could revise correspondingly.*

Thank you very much for pointing out this; we have revised this throughout the entire manuscript.

1.2 Results on conformation sampled via perturbing the torsion angle

▷ Review Comment: *Since most of the conformation variety of molecules comes from rotating the covalent bond*
*(i.e. the torsion angles), could the author provide results by randomly sampling alternative input structure via*
*perturbing the torsion angle. You can use the conformation sampling model like [1]. Could the authors provide*
*further results showing the behavior and robustness of Uni-Mol+ when inputting alternative conformations*
*based on the torsion angle perturbations?*

Thank you very much for your insightful suggestion. We have conducted further experiments to assess
the robustness of our model when employing conformations obtained through random perturbations
of torsion angles. Specifically, we applied Gaussian noise (with standard deviations of 0.005π , 0.01π
and 0.02π) to every torsion angle in the conformations generated by RDKit. As illustrated in Table
1, the results indicate that the input conformation with perturbing the torsion angle almost does not
affect Uni-Mol+’s performance. In other word, Uni-Mol+ is also robust to the torsion angle noises.
We have elaborated on these discussions in Supplementary Section 3.

Table 1: The benchmark results on PCQM4MV2, with different initial conformations.

Method	Valid MAE (\downarrow)
Uni-Mol+	0.0695
Uni-Mol+ w/ Noisy RDKit Conf., std=0.1	0.0695
Uni-Mol+ w/ Noisy RDKit Conf., std=0.3	0.0694
Uni-Mol+ w/ Torsion Angle Perturbed RDKit Conf., std= 0.005π	0.0698
Uni-Mol+ w/ Torsion Angle Perturbed RDKit Conf., std= 0.01π	0.0695
Uni-Mol+ w/ Torsion Angle Perturbed RDKit Conf., std= 0.02π	0.0696
Uni-Mol+ w/ 2D Conf.	0.0715

1.3 The Pre-training Potentials

▷ Review Comment: *Thank the authors for clarifying. Could the author further discuss the following situation*
*of Uni-Mol+: the SotA performance of Uni-Mol+ might rely on the fact that the involved two benchmark sets,*
*namely PCQM4MV2 and OC20, includes large-scale DFT equilibrium conformations as well as labels for*
*training. This is however not common to other real-world settings for molecular property prediction (PP) tasks,*
*where the data can be insufficient. On the other hand, the “pre-training” based models aim to induce fast and*
*effective domain adaptation via fine-tuning. In this sense, how do the authors see the potential applicability*
*of this Uni-Mol+ framework when (3D conf, label) data can be short, or provide evidence to justify the*
*performance of Uni-Mol+ in such a scenario?*

First and foremost, we would like to underscore the distinct nature of molecular property prediction
(PP) tasks as compared to quantum chemical (QC) property predictions, highlighting two primary
differences:

- • **Data Source Distinction:** PP tasks rely on data derived from wet lab experiments, which are
inherently resource-intensive and time-consuming. In contrast, QC properties are obtained
through electronic structure methods, which are more efficient and can generate large
datasets at a lower cost.
- • **Data Volume Discrepancy:** Due to the aforementioned reasons, the volume of data available
for QC is substantially larger than that for PP. Wet lab experiments, being costly and less
efficient, result in smaller datasets for PP, whereas computational methods can generate
extensive datasets for QC, as exemplified by databases such as OC20 and PCQM4MV2.

Given these differences, in the context of Uni-Mol+'s applications, the concern regarding "data can
be short" might be unwarranted.

Nevertheless, we acknowledge the importance of addressing scenarios where data may be limited.
To this end, we can use the pre-training strategy similar to those employed in PP tasks. This would
involve constructing a large dataset comprising equilibrium conformations (with QC labels being
optional) from sources like OC20, PCQM4Mv2, the Materials Project [1], or even datasets generated
in-house (akin to DeepMind's GNoME [2]). Then, we can apply Uni-Mol+'s methodology for
pre-training on this dataset. Subsequently, the pre-trained model could be fine-tuned on specific tasks
where data is limited, thus enhancing its applicability and performance across a broader range of
scenarios.

[1] Jain, A., Ong, S. P., Hautier, G., Chen, W., Richards, W. D., Dacek, S., Cholia, S., Gunter, D.,
Skinner, D., Ceder, G., & Persson, K. A. (2013). The Materials Project: A materials genome approach
to accelerating materials innovation. *APL Materials*, 1(1), 011002. <https://doi.org/10.1063/1.4812323>

[2] Merchant, A., Batzner, S., Schoenholz, S. S., Aykol, M., Cheon, G., & Cubuk, E. D. (2023).
Scaling deep learning for materials discovery. *Nature*, 624(7990), 80-85.

**2 Response to Reviewer #2**

We greatly appreciate the reviewer's additional comments. Below is our response.

▷ Review Comment: *Thank you for the additional results and the comprehensive analysis. My concerns are*
*mostly addressed in the revised manuscript except for one remaining request regarding "the effect of increasing*
*R on molecules grouped by different sizes". I appreciate the authors' study on the relationship between iterations*
*"R" and computational cost (time). To study the effect, I suggest the authors also include the "error" dimension.*
*For example, will larger molecules requires significantly more iterations to achieve a comparable error-level to*
*DFT? This can be done by performing some per-size-group evaluations to see if, for larger molecules, there is a*
*larger performance gap between Uni-Mol+ (with default R<=2) and DFT, and hence greater R is necessary.*
*This could complete the scalability analysis and help the community to understand the limitation of deep*
*learning-based approaches (if any).*

Thank you for your insightful feedback. Following your suggestion, we've extended our analysis to
explore how iterations ("R") and molecule sizes influence error metrics, presented in Figure 1 and
Table 2. This analysis revealed a notable trend: larger molecules tend to incur higher errors compared

Figure 1: The prediction error of increasing R on molecules grouped by different sizes.

to smaller molecules. Interestingly, our data indicates that increasing "R" does not significantly
 benefit larger molecules, as the performance at R=1 and R=2 is nearly identical.

Upon further examination, we recognized that the amount of larger molecules is quite small in
 validation set, as shown in Table 2. This led us to investigate the training data's molecular size
 distribution, detailed in Table 3. Most molecules in the training set are within the size range of
 (10, 20], which correlates with where we observe the lowest validation MAE. This indicates the the
 higher errors on larger molecules is due to the distribution of training data. This could be further
 improved by a training dataset with a broader distribution of molecular sizes, or models that have
 better generalizability on molecule sizes. We are very grateful for the reviewer's comment, which
 have helped us find this problem. We will leave it to future work. We have elaborated on these
 discussions in Supplementary Section 3.

Table 2: The prediction error of increasing R on molecules grouped by different sizes.

molecular size	num	R	Valid MAE (↓)
10	10404	1	0.0767
		2	0.0787
20	59865	1	0.0664
		2	0.0660
30	3146	1	0.1097
		2	0.1096
40	106	1	0.1622
		2	0.1543
50	23	1	0.1769
		2	0.1789

Table 3: Data distribution of Training set.

molecular size	num
10	319135
20	3059203

REVIEWERS' COMMENTS

Reviewer #2 (Remarks to the Author):

Thank you for the additional results and discussion. I have no further concerns.